# LapSum – One Method to Differentiate Them All: Ranking, Sorting and Top-k Selection

**Łukasz Struski** [1]   **Michał B. Bednarczyk** [1 2]   **Igor T. Podolak** [1]   **Jacek Tabor** [1]

## Abstract

We present a novel technique for constructing differentiable order-type operations, including soft ranking, soft top-k selection, and soft permutations. Our approach leverages an efficient closed-form formula for the inverse of a function LapSum, defined as a sum of Laplace distributions. This formulation ensures low computational and memory complexity in selecting the highest activations, enabling losses and gradients to be computed in $O(n \log n)$ time. Moreover, LapSum can easily be parallelized, both with respect to time and memory. Through extensive experiments, we demonstrate that our method outperforms state-of-the-art techniques for high-dimensional vectors and large $k$ values. Furthermore, we provide efficient implementations for both CPU and CUDA environments, underscoring the practicality and scalability of our method for large-scale ranking and differentiable ordering problems.

## 1. Introduction

Neural networks are trained using data through gradient descent, which requires models to be differentiable. However, common ordering tasks such as sorting, ranking, or top-k selection are inherently non-differentiable due to their piecewise constant nature. This typically prevents the direct application of gradient descent, which is essential for efficient learning from data. These challenges have become increasingly relevant in recent years (Lapin et al., 2016; Blondel et al., 2020; Petersen et al., 2022b). To address

---
[*]Equal contribution   [1]Faculty of Mathematics and Computer Science, Jagiellonian University, Kraków, Poland [2]University of Illinois, Urbana-Champaign, USA. Correspondence to: Łukasz Struski <lukasz.struski@uj.edu.pl>, Michał B. Bednarczyk <michalb3@illinois.edu>, Igor T. Podolak <igor.podolak@uj.edu.pl>, Jacek Tabor <jacek.tabor@uj.edu.pl>.

*Proceedings of the 42nd International Conference on Machine Learning*, Vancouver, Canada. PMLR 267, 2025. Copyright 2025 by the author(s).

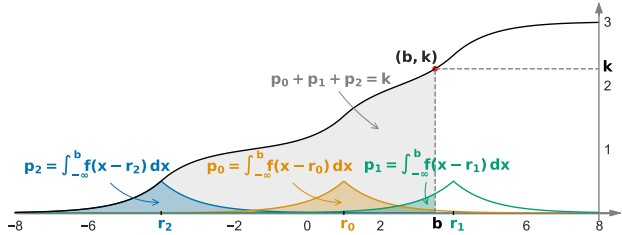

*Figure 1.* Scheme depicting the procedure for calculating the probabilities $p_i$ in the soft top-k selection algorithm for LapSum, given $n$ centers of Laplace distributions $r_i$ and a value $k < n$. All the formulae have a closed form and can be computed, jointly with derivatives, in $O(n \log n)$ time and $O(n)$ memory.

them, relaxations or approximations are needed to make such tasks compatible with neural network training. This can be achieved by smoothing the objective functions, introducing uncertainty into algorithms, softening constraints, or adding variability. The known methods include smoothed approximations (Berrada et al., 2018; Garcin et al., 2022), optimal based (Cuturi et al., 2019; Xie et al., 2020), permutations (Petersen et al., 2022b). Some of these solutions are often not closed form.

Challenges in addressing order related tasks arise from several factors. Among them are abrupt changes and instabilities when traversing regions. Time and memory issues often limit the solutions for large $n$ and $k$ values, where the aim lies in computing soft differenitiable top-k for large sequences of size $n$. There might also issues with scalability, parallel and batch processing. Techniques employed to solve these problems include relaxations and estimators, ranking regularizers, even learning-based ranking solutions. However, some problems remain unsolved or the existing methods are incomplete. E.g., efficiency in terms of time and memory usage may still be inadequate, impeding the solution of certain tasks (Xie et al., 2020). Frequently, there is a speed-precision trade-off. Several methods lack closed-form solutions, confining one to approximations or complicated calculations. Some methods do not yield probabilities (Blondel et al., 2020). Several do not yet support easy-to-use GPU methods.

Our aim was to address the above-mentioned issues. The main achievement of this paper is the construction of a general theory, based on arbitrary density, such that incorporation of the Laplace distribution provides a closed-form, numerically stable, and efficient solutions to all previously mentioned problems. More precisely, our approach, called LapSum, is based on a sum of Laplace distributions, see the scheme Fig. 1 for construction of soft top-k selection problem. We show that this approach gives a solution that is both theoretically sound and practical: the reduced $O(n \log n)$ time complexity is on par with that of sequence sorting, while the memory requirements are very limited in comparison to other available top-k solutions.

As the main contributions of this paper, we

- propose a novel, theoretically sound simple closed-form solution, called LapSum, for all classical soft-order based problems[1],

- prove and experimentally verify that LapSum has $O(n \log n)$ low time and $O(n)$ memory complexities, together with derivatives, outperforming within this aspect all other existing methods, see Fig. 2 for comparison with other SOTA methods,

- offer easy to use code for LapSum, for both CPU and CUDA, which makes our approach feasible for large optimization problems thanks to efficient use of parallelization.[2]

## 2. Related Work

The strategy of direct **top-k training** to optimize network performance, rather than straightforward use of top-1, was first introduced in fast top-k SVM (Lapin et al., 2016). The proposition called for the construction of differentiable loss functions to enable gradient learning.

Several methodologies are available. Some focus on **smooth approximations**. Berrada et al. (2018) proposed a **smooth top-k** to more accurately approximate the top-k loss and facilitate a direct technique for top-k gradient learning. Although these and comparable methods for direct gradient-based top-k learning yield precise results, they can be computationally intensive (Garcin et al., 2022). Some smooth approximations can speed up optimization, but may sacrifice accuracy.

Multiple methods employ the theoretically sound **optimal transport** framework, known for delivering precise gradients (Cuturi et al., 2019; Xie et al., 2020; Masud et al., 2023). However, these techniques might become computationally intensive when applied to large-scale tasks.

---

[1]Our basic top-k solution's pseudo-code is just 26 lines long (see Appendix A).

[2]Full code is available at github.com/gmum/LapSum.

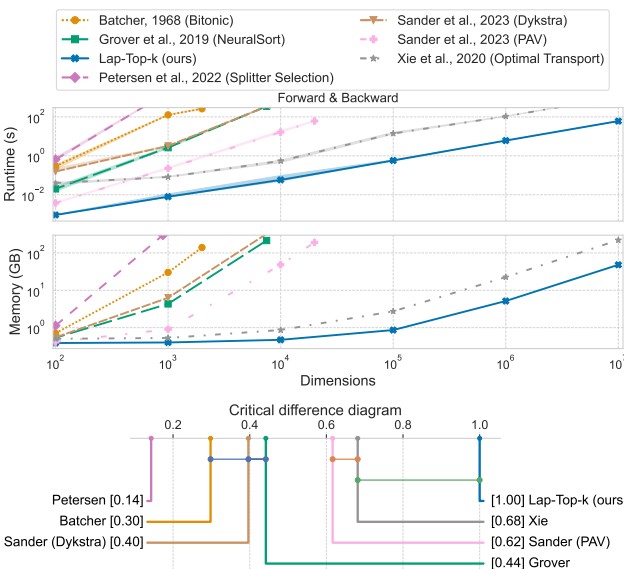

*Figure 2.* Upper: the relationship between the data dimension $n$ (horizontal axis) and the maximum memory usage and computation time (vertical axis), with $k = n/2$ in this figure. Computations for forward/backward processes shown here were performed on a CPU (see Appendix Figs. 10 to 13 for more). Below: the critical confidence diagram showing the statistical confidence of the above results (Demšar, 2006). LapSum approach is comparable to the best few (better on the right, statistically comparable joined with horizontal segments).

Various studies have expanded on these foundational concepts. **Ranking and learning to rank** techniques have evolved to deal with large-scale challenges, such as high dimensionality and handling partial derivatives (Xie et al., 2020; Zhou et al., 2021). Significant advances have been made in the proposal of methods that train efficiently with multiple labels through **permutation training** (Blondel et al., 2020; Petersen et al., 2022a;b; Shvetsova et al., 2023), sorting methods (Prillo & Eisenschlos, 2020; Petersen et al., 2022a), and the development and extraction of efficient **sparse networks** for use in expert systems (Sander et al., 2023). These approaches aim to improve computational efficiency, scalability, applicability to deep architectures, and effective hardware implementation.

Multiple studies demonstrate **effective applications** of top-k learning techniques in pattern recognition and feature extraction, the creation of sparse networks and feature extraction, and recommender systems, such as through the use of differentiable logic gates (Zhao et al., 2019; Hoefler et al., 2021; Chen et al., 2023; Xu et al., 2023; Chen et al., 2021).

Theoretical studies establish links between top-k learning and classical statistical learning, the margin methodology, and a broader framework, offering strategies to optimize the

k value Cortes et al. (2024); Mao et al. (2024).

## 3. General soft-order theory based on sums of cumulative density functions

We begin by demonstrating how a cumulative density function can be used to construct soft analogues of hard, order-based problems such as top-k selection and permutations. Although the proposed approach is primarily theoretical and not computationally efficient for a general density, in the next section we show that it can be efficiently computed with complexity $O(n \log n)$ in the specific case of the Laplace distribution.

### 3.1. Function $F$-Sum

We base our construction on an arbitrary even and strictly positive density function $f$, with $F$ denoting its cumulative density function, so that $F(-x) = 1 - F(x)$ and consequently $F(0) = 1/2$. Additionally, for a scale parameter $\alpha \neq 0$ we put

$$F_\alpha(x) = F\left(\frac{x}{\alpha}\right).$$

Observe that $F_\alpha(x) \to H(x)$ as $\alpha \to 0^+$, where $H$ denotes the Heaviside step function, with $H(0) = \frac{1}{2}$, and $F_\alpha(x) \to H(-x)$ as $\alpha \to 0^-$.

Given a sequence $(r_0, \ldots, r_{n-1}) \subset \mathbb{R}$, we consider the function

$$F\text{-Sum}_\alpha(r, x) = \sum_i F_\alpha(x - r_i).$$

Clearly, the image of $F$-$\text{Sum}_\alpha$ is the interval $(0, n)$. It occurs that $F$-$\text{Sum}_\alpha(r, x)$ is invertible as a function of $x$. This follows from the fact that since $f$ is a strictly positive density, $F_\alpha$ is a strictly increasing continuous function for $\alpha > 0$ and strictly decreasing for $\alpha < 0$. Consequently, $F$-$\text{Sum}_\alpha(x, r)$ is a continuous strictly monotonous function, which is concluded in the following observation.

**Observation 3.1.** *Let sequence $r = (r_0, \ldots, r_{n-1}) \in \mathbb{R}^n$ be given, and let $k \in (0, n), \alpha \neq 0$ be arbitrary. Then the equation*

$$F\text{-Sum}_\alpha(r, x) = k$$

*has a unique solution $x$, which we denote by $x = F\text{-Sum}^{-1}(k)$*

It should be noted, that in our reasoning we do not require $k$ to be integer.

**Derivatives** Note that the above functions are differentiable and that the derivatives can be computed efficiently. By $f_\alpha$ we denote the function $f_\alpha(x) = \frac{1}{\alpha} f(x/\alpha)$. Observe that, for $\alpha > 0$, it is the density whose CDF is equal to $F_\alpha$.

Then we have

$$\frac{\partial}{\partial x} F\text{-Sum}_\alpha(r, x) = \sum_i f_\alpha(x - r_i). \qquad (1)$$

Now we compute the derivatives of the inverse function $k \to F\text{-Sum}_\alpha^{-1}(r, k)$. Given $k \in (0, n)$, let $b = b_\alpha(r, k)$ denote the unique solution to

$$F\text{-Sum}(b) = \sum_i F_\alpha(b - r_i) = k.$$

To compute the derivative of the function $b$ with respect to $w$, we differentiate the above formula with respect to $k$ obtaining $\frac{\partial b}{\partial k} \sum_i f_\alpha(b - r_i) = 1$. Thus

$$\frac{\partial}{\partial k} F\text{-Sum}_\alpha^{-1}(r, k) = \frac{\partial b}{\partial k} = \frac{1}{\sum_i f_\alpha(b - r_i)}. \qquad (2)$$

### 3.2. Soft rankings and soft orderings

We show how $F$-Sum may be applied to provide solutions to differentiable soft order problems. Let us start with the soft rankings problem.

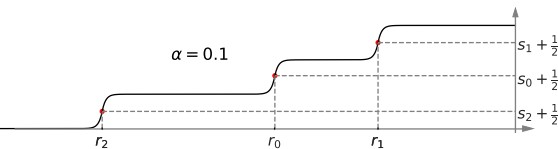

*Figure 3.* The computation of the soft ranking based on $F$ given as CDF of Laplace distribution, here $(s_0, s_1, s_2) = F\text{-Rank}_{\alpha=1}(r_0, r_1, r_2)$.

**Soft rankings** For $\alpha \neq 0$, fixed sequence $(r_i)$ and an arbitrary $j \in \{0, \ldots, n - 1\}$ we define

$$F\text{-Rank}_\alpha(r_j) = F\text{-Sum}_\alpha(r, r_j) - \frac{1}{2},$$

see Fig. 3. By a soft ranking of whole sequence $r = (r_i)$ we understand

$$F\text{-Rank}_\alpha(r) = (F\text{-Rank}_\alpha(r_i))_i.$$

First, we show that $F\text{-Rank}_\alpha(r_j) \subset (0, n-1)$. This follows directly from the formula

$$F\text{-Rank}_\alpha(r_j) = \sum_{l \neq j} F_\alpha(r_j - r_l), \qquad (3)$$

which is a consequence of the equality $F_\alpha(r_j - r_j) = F(0) = \frac{1}{2}$.

One can easily observe that $F\text{-Rank}_\alpha$ is permutation-equivariant, that is, a permutation of the input will result in a similar permutation of the output. By $s_i^+$ we denote the rank of $r_i$ in sequence $r$ (with respect to standard order), i.e.

$$s_i^+ = \text{card}\{j : r_j < r_i\} = \sum_{j \neq i} H(r_i - r_j).$$

With $s_i^-$ we denote the rank of element $r_i$ in the reverse order of $r$, that is, $s_i^- = (n-1) - s_i^+$. To prove that $F\text{-Rank}_\alpha$ is a correct soft version of ranking operation, we need to show that in the limiting case $\alpha \to 0^\pm$ we obtain the hard ranking.

**Theorem 3.2.** *For a sequence $r = (r_i)$ of pairwise distinct elements we have*

$$F\text{-Rank}_\alpha(r_j) \to s_j^\pm \text{ as } \alpha \to 0^\pm,$$

*for arbitrary $j \in \{0, \ldots, n-1\}$.*

*Proof.* Let us prove the limiting formula above for $\alpha > 0$ (the case $\alpha < 0$ is similar). Since $F_\alpha$, for $\alpha \to 0^+$, converges to a Heaviside function, by (3) we get

$$\lim_{\alpha \to 0^+} F\text{-Rank}_\alpha(r_j) = \lim_{\alpha \to 0^+} \sum_{i \neq j} F_\alpha(r_j - r_i)$$
$$= \sum_{i \neq j} H(r_j - r_i) = s_j^+. \quad \square$$

Thus, with $\alpha \to 0^+$, we obtain the hard ranking of the sequence $r$ in increasing order, while for $\alpha \to 0^-$ we obtain the ranking in decreasing order.

**Soft orderings** Soft ordering can be intuitively seen as an inverse operator to soft ranking. Given a sequence $(r_i)$ we define

$$(F\text{-Sort}_\alpha(r))_l = F\text{-Sum}_\alpha^{-1}(\tfrac{1}{2} + l) \text{ for } l \in \{0, \ldots, n-1\}.$$

Informally we can write

$$F\text{-Sort}_\alpha(r) = F\text{-Sum}_\alpha^{-1}\left(\tfrac{1}{2} + (0, 1, \ldots, n-1)\right).$$

It can now be readily demonstrated that for a sequence $r$ with distinct elements, the aforementioned operation has the following properties, see Fig. 4:

- it is permutation invariant,
- $F\text{-Sort}_\alpha(r)$ converges to the sorted (with respect to increasing order) $r$ for $\alpha \to 0^+$,
- $F\text{-Sort}_\alpha(r)$ converges to sorted $r$ (with respect to decreasing order) as $\alpha \to 0^-$.

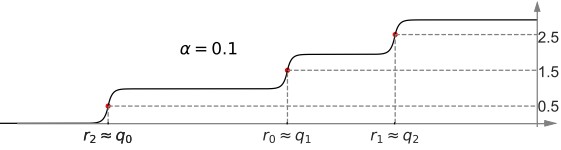

*Figure 4.* The result of soft sorting with $F$ given by CDF of Laplace distribution. Observe that for small $\alpha$ the constructed sequence $(q_0, q_1, q_2) = F\text{-Sort}_{\alpha=1}(r_0, r_1, r_2)$ practically coincides with the sorted sequence $r$.

### 3.3. Soft top-k

We proceed to the construction of soft differentiable version of top-k selection problem. The aim is to construct a differentiable version of the hard top-k (formally max top-k) operation which, for a sequence $r = (r_i) \in \mathbb{R}^n$, returns a binary sequence $(p_i) \in \{0, 1\}^n$ that has a value 1 for the indices with k highest values of $r$, and zero for all other indices. Clearly, $\sum_i p_i = k$.

In our approach, we can choose an arbitrary $k \in (0, n)$, where we underline that we do not require $k$ to be an integer. Using $F\text{-Sum}$ we construct the parameterization of the space

$$\Delta_k = \{[p_0, \ldots, p_{n-1}] \in (0, 1)^n : \sum_i p_i = k\}.$$

Formally, we define

$$F\text{-Top}_\alpha(r, k) = (F_\alpha(b - r_0), \ldots, F_\alpha(b - r_{n-1}))$$
$$\text{where } b = F\text{-Sum}_\alpha^{-1}(r, k),$$

see Fig. 1 for a visualization of this approach.

We show that this gives us the correct solution for the soft top-k problem.

**Theorem 3.3.** *For every $k \in (0, n)$ and $r \in \mathbb{R}^n$ we have*

$$F\text{-Top}_\alpha(r, k) \in \Delta_k. \tag{4}$$

*Moreover, if $k$ is integer and the elements of $r$ are pairwise distinct, then*

- *$F\text{-Top}_\alpha(r, k) \to \text{top min}_k(r) \text{ as } \alpha \to 0^+$,*
- *$F\text{-Top}_\alpha(r, k) \to \text{top max}_k(r) \text{ as } \alpha \to 0^-$.*

*Proof.* Observe that (4) follows directly from the fact that $b = F\text{-Sum}_\alpha^{-1}(r, k)$, and thus $\sum_i F_\alpha(b - r_i) = k$.

Due to the limited space, the proof of the limiting case is given in the Appendix Theorem B.1. $\square$

**Invertibility of $F$-Sum** One can observe that the function $F\text{-Top}$ has properties similar to that of softmax. In particular, if two outputs are equal, then the inputs are equal modulo some constant translation. Namely, by the invertibility of $F$ we obtain that if $F\text{-Top}_\alpha(r, k) = F\text{-Top}_\alpha(\bar{r}, k)$ then

$$\bar{r}_i = r_i + (\bar{b} - b) \text{ for every } i,$$

where $b = F\text{-Sum}_\alpha^{-1}(r, k)$, $\bar{b} = F\text{-Sum}_\alpha^{-1}(\bar{r}, k)$. This implies, in particular, that similarly to softmax, $F\text{-Top}_\alpha$ with domain restricted to $\{r = (r_i) \in \mathbb{R}^n : \sum_i r_i = 0\}$ is invertible.

### 3.4. Soft permutations

The task of the soft permutation problem is to generalize the hard permutation matrix to a differentiable double stochastic

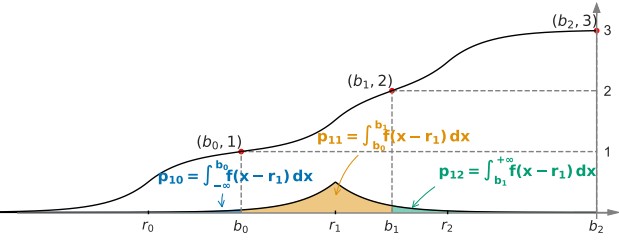

*Figure 5.* Illustration of probability calculation for sort/permutation method. We show the underlying processes for generating probabilities under given conditions.

matrix.

Recall that the *permutation matrix* corresponding to a permutation $(s_i)_{i=1..n}$ of the indices $(1, \ldots, n)$ is a square binary matrix that represents the reordering of elements in the sequence. Precisely, the permutation matrix $P$ is a matrix $n \times n$ where:

- each row and column contains exactly one entry of $1$, and all other entries are $0$,

- if $P$ is applied to a vector $v$, the result is a reordering of $v$ according to $(s_i)$.

To construct $P$, first create a zero matrix $n \times n$, and then, for each $i \in \{1, 2, \ldots, n\}$, place a $1$ in the $(i, p_i)$-th position.

**Example 3.4.** *Let* $(s_i) = (3, 1, 2)$. *The permutation matrix is:* $P = \begin{bmatrix} 0 & 0 & 1 \\ 1 & 0 & 0 \\ 0 & 1 & 0 \end{bmatrix}$.

For a sequence $(r_i)$ of pairwise distinct scalars, by its (hard) permutation matrix, we understand the permutation matrix of $(s_i)$, where $s_i$ is the rank order of $r_i$ in the ordered $r$.

By $\mathrm{B}_n$ we denote the space of all doubly stochastic matrices of size $n \times n$, where we recall that a doubly stochastic matrix is a square matrix of non-negative real numbers where each row and each column sum up to 1.

**Soft permutation task**   The aim of soft permutation task is to define a differentiable function $\mathcal{P} : \mathbb{R}^n \to M_{n \times n}$, so that $\mathcal{P}(r) \in \mathrm{B}_n$ for $r \in \mathbb{R}^n$ and that hard permutation can be obtained as a limiting case of soft permutation. We consider only $\alpha > 0$.

**Definition 3.5.** *Given* $r = (r_0, \ldots, r_{n-1}) \in \mathbb{R}^n$ *and* $\alpha > 0$ *we define*

$$F\text{-Perm}_\alpha(r) = [F_\alpha(b_{i+1} - r_j) - F_\alpha(b_i - r_j)]_{i,j},$$

*where* $b_{-1} = -\infty$, $b_i = F^{-1}(i)$ *for* $i = 0, \ldots, n-1$, $b_n = \infty$.

Visualization of the definition is given in Fig. 5. We proceed to theorem, which gives justification of the defined function.

**Theorem 3.6.** *Let* $r \in \mathbb{R}^n$ *be arbitrary and let* $\alpha > 0$. *Then*

$$F\text{-Perm}_\alpha(r) \in \mathrm{B}_n.$$

*Moreover, if* $r$ *has pairwise distinct elements, then*

$$F\text{-Perm}_\alpha(r) \to permutation\_matrix(r) \text{ as } \alpha \to 0^+.$$

*Proof.* We prove that $F\text{-Perm}_\alpha(r)$ is doubly stochastic. Let an arbitrary $j \in \{0, \ldots, n-1\}$. Then

$$\sum_{i=0}^{n-1} (F_\alpha(b_{i+1} - r_j) - F_\alpha(b_i - r_j))$$

$$= F_\alpha(\infty) - F_\alpha(-\infty) = 1.$$

Now fix an arbitrary $i \in \{0, \ldots, n-1\}$. Then

$$\sum_{j=0}^{n-1} (F_\alpha(b_{i+1} - r_j) - F_\alpha(b_i - r_j)) =$$

$$= F\text{-Sum}(r, b_{j+1}) - F\text{-Sum}(r, b_j) = (j+1) - j = 1.$$

The proof of the limiting case $\alpha \to 0^+$ can be done similarly to the reasoning from Theorem B.1 in the Appendix. $\square$

# 4. Theory behind LapSum

In the previous section we have constructed a general soft-order theory based on an arbitrary cumulative density function $F$. However, such an approach is in practice unapplicable, since the complexity of evaluating $F\text{-Sum}(r, x)$ for a sequence $r$ of size $n$ is in general quadratic. Moreover, to efficiently apply the theory, we need to be able to compute the inverse function $F\text{-Sum}^{-1}(r, w)$. Finally, to use the model for neural networks, we would optimally need the derivatives computed in complexity $O(n \log n)$.

## 4.1. Where the magic happens ...

We demonstrate that all the above mentioned problems can be solved when $F$ represents the CDF of the standard Laplace distribution given by

$$Lap(x) = \begin{cases} \frac{1}{2} \exp(x) & \text{for } x \leq 0, \\ 1 - \frac{1}{2} \exp(-x) & \text{for } x > 0. \end{cases}$$

The crucial role in our experiments will be played by the function $Lap\text{-Sum}$, which is the sum of the Laplace cumulative density functions. Recall that for sequence $r = (r_i)_{i=0..n-1} \subset \mathbb{R}$, applying the definition for $F\text{-Sum}$ for $F = Lap$ we get

$$Lap\text{-Sum}_\alpha(r, x) = \sum_{i=0}^{n-1} Lap_\alpha(x - r_i),$$

where $Lap_\alpha(x) = Lap(x/\alpha)$.

The function $Lap\text{-Sum}$ will be crucial in solving all previ-

ously mentioned soft-order problems – we shall denote the family of methods constructed with the use of $Lap$-Sum function as LapSum. We show two crucial facts which enable the efficient use of LapSum in soft-order problems:

- we can evaluate $Lap$-Sum at $n$ points in complexity $O(n \log n)$,
- the computation of $Lap$-Sum can be parallelized,
- the function $Lap$-Sum has inverse given by a closed-form formula.

**Theorem 4.1.** *Suppose that $r = (r_i)_{i=0..n-1} \subset \mathbb{R}$ is an increasing sequence. We can compute the values of $Lap$-Sum for an increasing sequence $x = (x_j)_{j=0..m-1} \subset \mathbb{R}$ with a complexity of $O(n + m)$.*

To prove this theorem, we will need the following proposition.

**Proposition 4.2.** *Assume that $(r_i)_{i=0..n-1}$ is an increasing sequence and $\alpha > 0$, where we additionally put $r_{-1} = -\infty, r_n = \infty$.[3]*

*Then*

$$Lap\text{-Sum}(x) =$$
$$= \tfrac{1}{2}a_j \exp\left(\tfrac{x - r_{j+1}}{\alpha}\right) - \tfrac{1}{2}b_{j+1} \exp\left(\tfrac{r_j - x}{\alpha}\right) + c_{j+1}$$
$$\text{for } x \in [r_j, r_{j+1}], j = -1..(n-1).$$
$$(5)$$

*where sequences $(a_i)_{i=-1..n-1}, (b_i)_{i=0..n} \subset \mathbb{R}$ and $(c_i)_{i=0..n} \subset \mathbb{N}$ are defined iteratively by the formulae*

- $a_{N-1} = 0, \; a_{k-1} = \exp(\tfrac{r_{k-1} - r_j}{\alpha}) \cdot (1 + a_j),$
- $b_0 = 0, \qquad b_{j+1} = \exp(\tfrac{r_j - r_{j+1}}{\alpha}) \cdot (1 + b_j),$
- $c_0 = 0, \qquad c_{j+1} = 1 + c_j.$

*Proof.* Given that $F$ on the intervals $(-\infty, 0]$ and $[0, \infty)$ is composed of constants, $\exp(x/\alpha)$, and $\exp(-x/\alpha)$, this property extends to function $Lap\text{-Sum}_\alpha(x; r)$ for every interval $[r_j, r_{j+1}]$. Consequently, there exist sequences $a_j, b_j, c_j$ that satisfy formula (5). We aim to demonstrate that the proposition's recurrent formulas are valid.

Consider the interval $(-\infty, r_0)$. Clearly, $c_0$ is zero because the expression for $Lap$-Sum in this range is composed of exponentials. Assuming that the expression for $c_j$ holds, it follows that in the next interval, we increase by 1. This logic is uniformly applied to all subsequent $c_j$. Now, consider the sequence $a_j$. Initially, note that $a_{n-1} = 0$. Using a similar reasoning, we derive the entire recursive formula. $\square$

We are now ready to present the proof of Theorem 4.1.

*Proof of Theorem 4.1.* Since the sequence $(r_i)_{i=0..n-1}$ is ordered, then, according to the above proposition, we can

---

[3]The case for $\alpha < 0$ can be treated analogously.

compute the sequences $a_i, b_i, c_i$ in complexity $O(n)$. Now in complexity $O(n+m)$ we can find $n_l$ for $l = 0, \dots, j-1$ such that $x_l \in [r_{n_l-1}, r_{n_l}]$. Finally, thanks to the proposition, we obtain $Lap\text{-Sum}_\alpha(x_l)$ is given by

$$\tfrac{1}{2}a_j \exp(\tfrac{x_l - r_{j+1}}{\alpha}) - \tfrac{1}{2}b_{j+1} \exp(\tfrac{r_j - x_l}{\alpha}) + c_{j+1},$$

which directly implies that we can compute all these values in $O(n + m)$ complexity. $\square$

**Parallelization**   Observe that the iterative formulas for $a_i, b_i$ and $c_i$ can be computed in parallel by applying the standard prefix scan approach.

**Calculation of inverse function**   We now show that function $(0, n) \ni w \to Lap\text{-Sum}_\alpha^{-1}(r, k)$ can be computed with a simple straightforward formula. For clarity, we present the formulae for $\alpha = 1$.

First compute $w_i = Lap\text{-Sum}(r_i)$ for all $i = 0..n - 1$. If $k \in (-\infty, r_0]$, then as on this interval $Lap\text{-Sum}(x) = \tfrac{a_0}{2} \exp(x - r_0)$, solution to $Lap\text{-Sum}(x) = k$ is given by

$$x = r_0 + \log 2 + \log w - \log a_0$$

If $k \in [r_{i-1}, r_i]$ for some $0 < i < n$ then, since

$$Lap\text{-Sum}(x) = \tfrac{1}{2}a_j \exp(x - r_{j+1})$$
$$- \tfrac{1}{2}b_{j+1} \exp(r_j - x) + c_{j+1},$$

by direct calculations, the solution for $Lap\text{-Sum}(x) = k$ is

$$x = r_{j+1} - \log a_j +$$
$$+ \log(k - c_{j+1} + \sqrt{|k - c_{j+1}|^2 + a_j b_{j+1} \cdot e^{r_j - r_{j+1}}}).$$

If $k \in [r_n, \infty)$, we analogously compute

$$x = r_{n-1} - \log 2 - \log(c_n - k) + \log b_n.$$

### 4.2. Efficient computation of derivatives

The fascinating aspect of the functions being analyzed is our ability to calculate all required derivatives in $O(n \log n)$ time. The key to this efficiency is that instead of computing derivatives in matrix form, we construct formulas for right and left multiplication of the derivative by row and column vectors, respectively. A more detailed explanation is provided in the Appendix.

Let us start with the case of top-k selection, namely for function $Lap\text{-Top}_\alpha(r, k)$. Define the function $P : \mathbb{R}^n \ni r = (r_i) \to p = (p_i) \in \Delta_k \subset [0, 1]^n$, where

$$p_i = Lap_\alpha(b - r_i) \text{ where } b = Lap\text{-Sum}_\alpha^{-1}(r, k).$$

It can be shown (see Appendix C.2), that the derivative of $p$ with respect to $r$ is given by the matrix

$$D = \frac{\partial P}{\partial w} = s\, q^T - \text{diag}(s), \quad (6)$$

where $s = (s_i) \in \mathbb{R}^n$ is given by $s_i = \tfrac{1}{\alpha} \min(p_i, 1 - p_i)$

and

$$q = \text{softmax}\left(-\left|\tfrac{b-r_0}{\alpha}\right|, \ldots, -\left|\tfrac{b-r_{N-1}}{\alpha}\right|\right).$$

For optimizing purposes, utilizing such a constructed derivative would result in a complexity of at least $O(n^2)$. However, note that to apply gradient optimization it is actually enough to compute $v^T D$ and $D v$ for an arbitrary vector $v$. By executing direct calculations, one can trivially derive from (6) the formulae

$$D v = \langle q, v \rangle s - s \odot v, \text{ and } v^T D = \langle s, v \rangle q^T - q^T \odot v^T,$$

where $\odot$ denotes component-wise multiplication.

Another important function, necessary to calculate soft-sorting and permutations is, for fixed $k \leq n$, the function

$$L(r, k_0, \ldots, k_{j-1}) =$$
$$= (Lap\text{-Sum}_\alpha^{-1}(r, k_0), \ldots, Lap\text{-Sum}_\alpha^{-1}(r, k_{j-1})).$$

In this instance, calculating the derivative of $L$ efficiently, in particular the left and right multiplication of the derivative by row and column vectors, respectively, can also be implemented within a time complexity of $O(n \log n)$.

## 5. Experiments and results

The main incentive of this paper was to design a new and simple to implement closed-form method that would constitute a tool for use in all the order tasks. We wanted LapSum to outperform other methods in terms of time and memory complexity, while obtaining the general SOTA in the standard experiments.

### 5.1. Experiments on soft-top-k methods

Our evaluation consists of training from scratch on CIFAR-100 and fine-tuning on ImageNet-1K and ImageNet-21K-P datasets. These evaluations aim to assess LapSum's influence of top-1 and top-5 training on accuracy metrics. The experiment highlights the effect of the smaller number of classes in CIFAR-100 compared to the larger count of 1000 in ImageNet-1K and an even greater count of 10450 in ImageNet-21K-P. We employ $P_j$ settings akin to those in Petersen et al. (2022b). Experiment details are available in Appendix D.

Refer to Tab. 1 for the CIFAR-100 results, where we used $P_j = [0., 0., 0., 0., 1.]$ for top-5 learning. Furthermore, we used $[.2, .2, .2, .2, .2]$ to have a direct comparison with the results given in Petersen et al. (2022b). Our experiments demonstrated superior or at least comparable results. The results are consistently stable and are fast to obtain.

Tab. 2 summarizes the outcomes of fine-tuning on ImageNet-1K and ImageNet-21K-P, similarly to the setup in Petersen et al. (2022b). The ImageNet-21K-P results show LapSum surpass alternative methods. LapSum yields slightly su-

*Table 1.* Performance comparison of permutation-based methods on CIFAR-100 using ResNet18. The table shows ACC@1 (standard accuracy) and ACC@5 (top-5 accuracy). Bold and italic values denote the best and second-best results, respectively. $P_j$ represents the training probability distribution. Methods marked with (*) are based on Berrada et al. (2018).

| Method | $P_j$ | CIFAR-100 | |
|---|---|---|---|
| | | ACC@1 | ACC@5 |
| Smooth top-$k^*$ | [0.,0.,0.,0.,1.] | 53.07 | 85.23 |
| NeuralSort | [0.,0.,0.,0.,1.] | 22.58 | 84.41 |
| SoftSort | [0.,0.,0.,0.,1.] | 01.01 | 05.09 |
| SinkhornSort | [0.,0.,0.,0.,1.] | *55.62* | *87.04* |
| DiffSortNets | [0.,0.,0.,0.,1.] | 52.81 | 84.21 |
| Lap-Top-k (ours) | [0.,0.,0.,0.,1.] | **58.07** | **87.50** |
| NeuralSort | [.2,.2,.2,.2,.2] | 61.46 | 86.03 |
| SoftSort | [.2,.2,.2,.2,.2] | 61.53 | 82.39 |
| SinkhornSort | [.2,.2,.2,.2,.2] | 61.89 | *86.94* |
| DiffSortNets | [.2,.2,.2,.2,.2] | *62.00* | 86.73 |
| Lap-Top-k (ours) | [.2,.2,.2,.2,.2] | **64.53** | **88.51** |

perior results faster due to LapSum's efficient time and memory management in handling this intricate task. The results of ImageNet-1K are on par with those of other methods. This demonstrates that LapSum is particularly well suited for high-dimensional tasks, being fast and easy to use for other tasks. At the same time, LapSum has a differentiable parameter $\alpha$ (see Sec. 3.1), which can be used to fine-tune individual solutions. The results of these experiments are presented in Appendix D.

Petersen et al. (2022b) algorithm suggests a predetermined number of $m$ best results to compute for differential ranking. This increases its effectiveness.

**Runtime, and memory analysis** Fig. 2 depicts the experimental time–memory correlations for LapSum alongside other algorithms. We exclude (Blondel et al., 2020) as it does not return probability vectors (Petersen et al., 2022b); we also exclude (Prillo & Eisenschlos, 2020) as we found their method not working for some random input data. The relationships of time/memory against data dimension $n$ and $k$, with $k = n/2$, is considered. It is evident that, concerning memory usage, LapSum (the blue solid line) outperforms, or at least matches, the best competing approaches, in particular for higher data dimensions. Some algorithms could not complete tasks due to memory limitations. Similarly, this holds for time complexity as well. These patterns are consistent for both the standalone forward pass and the combined forward and backward passes, using either CPU or GPU (see also Figs. 10 to 13 in the Appendix). To assess the statistical significance of these experimental comparisons, we have performed a critical difference CD ranking test (Demšar, 2006), see Fig. 2, which shows that LapSum outperforms other competitive algorithms.

*Table 2.* Results on ImageNet-1K and ImageNet-21K-P for fine-tuning the head of ResNeXt-101 (Mahajan et al., 2018). We report ACC@1 and ACC@5 accuracy metrics, averaged over 10 seeds for ImageNet-1K and 2 seeds for ImageNet-21K-P. Methods are evaluated using different $P_j$ configurations, highlighting the effectiveness in optimizing ranking-based losses. Best performances are in bold, with second best in italics. Methods labeled with (*) are based on Berrada et al. (2018).

| Method | $P_j$ | ImgNet-1K | | ImgNet-21K-P | |
|---|---|---|---|---|---|
| | | ACC@1 | ACC@5 | ACC@1 | ACC@5 |
| Smooth top-$k^*$ | [0.,0,0,0,1.] | 85.15 | 97.54 | 34.03 | 65.56 |
| NeuralSort | [0.,0,0,0,1.] | 33.37 | 94.75 | 15.87 | 33.81 |
| SoftSort | [0.,0,0,0,1.] | 18.23 | 94.97 | 33.61 | *69.82* |
| SinkhornSort | [0.,0,0,0,1.] | **85.65** | **98.00** | *36.93* | 69.80 |
| DiffSortNets | [0.,0,0,0,1.] | 69.05 | 97.39 | 35.96 | 69.76 |
| Lap-Top-k (our) | [0.,0,0,0,1.] | *85.47* | 97.83 | **37.93** | **70.67** |
| NeuralSort | [.5,0,0,0,.5] | **86.30** | 97.90 | 37.85 | 68.08 |
| SoftSort | [.5,0,0,0,.5] | 86.26 | *97.96* | 39.93 | 70.63 |
| SinkhornSort | [.5,0,0,0,.5] | *86.29* | **97.97** | 39.85 | 70.56 |
| DiffSortNets | [.5,0,0,0,.5] | 86.24 | 97.94 | *40.22* | 70.88 |
| Lap-Top-k (our) | [.5,0,0,0,.5] | 86.28 | 97.93 | **40.48** | **71.05** |

*Table 3.* Accuracy of $k$NN-based methods on MNIST and CIFAR-10 comparing standard $k$NN with enhanced variants. Bold and italic values stand for best and second-best results. Our proposed $k$NN+Lap-Top-$k$ shows strong performance, especially on CIFAR-10, showing the benefits of Laplacian-based top-k optimization.

| Algorithm | MNIST | CIFAR-10 |
|---|---|---|
| $k$NN | 97.2 | 35.4 |
| $k$NN+PCA | 97.6 | 40.9 |
| $k$NN+AE | 97.6 | 44.2 |
| $k$NN+pretrained CNN | 98.4 | 91.1 |
| RelaxSubSample | 99.3 | 90.1 |
| $k$NN+NeuralSort | **99.5** | 90.7 |
| $k$NN+OT | 99.0 | 84.8 |
| $k$NN+Softmax $k$ times | 99.3 | *92.2* |
| CE+CNN | 99.0 | 91.3 |
| $k$NN+SOFT Top-$k$ | *99.4* | **92.6** |
| $k$NN+Lap-Top-$k$ (ours) | *99.4* | 92.2 |

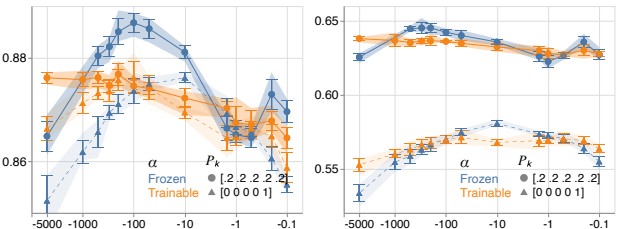

*Figure 6.* LapSum top-1 and top-5 accuracies across varying values of $\alpha$ and top-5 training on *CIFAR-100*. Higher $\alpha$ produces more discrete selections, lower $\alpha$ lead to smoother outputs, influencing how strictly the model enforces a top-k criterion.

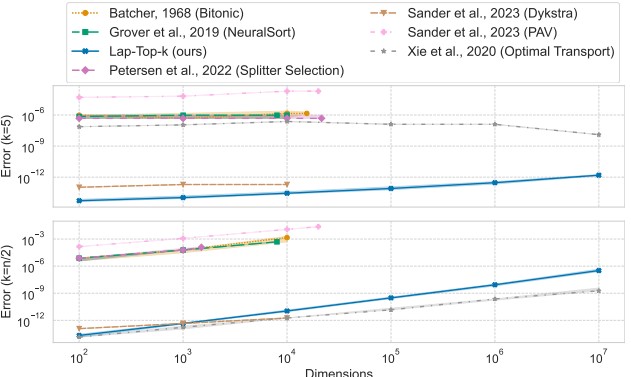

*Figure 7.* The error calculated as Error(k) $= |\sum_{i=1}^{n} p_i - k|$, where $p_i$, as a function of $k$ (the value defining Top-$k$ in the respective methods) and $n$ (the data dimension). The plots correspond to the errors obtained in the computations presented in Fig. 2.

Fig. 6 shows the impact of LapSum $\alpha$ trainable parameter value on training on the resulting in harder or softer solutions (see Appendix D, Fig. 8). We have also performed experiments on the accuracy of direct approximations of $k$ value using LapSum, see Fig. 7. Our proposition consistently gives better approximations than competing approaches, irrespective of a given problem dimension.

**$k$-NN for Image Classification**   The protocol followed was adapted from (Xie et al., 2020).[4] Tab. 3 shows test accuracies on MNIST and CIFAR-10. We see that Lap-top-k does not deteriorate the model performance. We provide additional details in Appendix D.

---

[4]We use the code kindly made available by authors.

### 5.2. Experiments on soft-permutation methods

To assess the effectiveness of soft-permutation techniques techniques, we use an enhanced version of the MNIST dataset, the large-MNIST (Grover et al., 2019) where four random MNIST digits are combined into one composite image representing four-digits number. This allows to evaluate the techniques' ability to learn correct permutations.

Baseline methods employ a common to all CNN to encode images into a feature space. The row-stochastic baseline combine CNN features into a vector processed by a multilayer perceptron for multiclass predictions. Sinkhorn and Gumbel-Sinkhorn approaches use the Sinkhorn operator to create doubly-stochastic matrices from these features. NeuralSort-based methods apply the NeuralSort operator with Gumbel noise to produce unimodal row-stochastic matrices. The loss in all methods is the row-wise cross-

*Table 4.* Mean permutation accuracy (%) on testset for Large-MNIST for different number of randomly sampled images $n$. First value is the fraction of permutations correctly identified, while the one in parentheses is the fraction of individual element ranks correctly predicted. Results for competing methods are from Grover et al. (2019). Bold and italic values stand for best and second-best results. GumbelS stands for Gumbel-Sinkhorn method, DNeural for deterministic NeuralSort, SNeural for Stochastic NeuralSort.

| Method | n = 3 | n = 5 | n = 7 | n = 9 | n = 15 |
|---|---|---|---|---|---|
| Vanilla | 46.7 (80.1) | 9.3 (60.3) | 0.9 (49.2) | 0.0 (11.3) | 0.0 (6.7) |
| Sinkhorn | 46.2 (56.1) | 3.8 (29.3) | 0.1 (19.7) | 0.0 (14.3) | 0.0 (7.8) |
| GumbelS | 48.4 (57.5) | 3.3 (29.5) | 0.1 (18.9) | 0.0 (14.6) | 0.0 (7.8) |
| DNeural | *93.0* (*95.1*) | *83.7* (*92.7*) | 73.8 (**90.9**) | **64.9** (**89.6**) | *38.6* (*85.7*) |
| SNeural | 92.7 (95.0) | 83.5 (92.6) | **74.1** (**90.9**) | *64.6* (*89.5*) | **41.8** (**86.2**) |
| Lap(ours) | **94.2** (**96.1**) | **85.3** (**93.2**) | **74.1** (*90.7*) | 63.1 (88.6) | 33.9 (82.4) |

entropy against ground-truth permutation matrices with row-stochastic outputs. In our method LapSum we apply the same backbone architecture while eliminating the need for additional layers[5] to transform the stochastic matrix. Tab. 4 provides results of this approach against other methods.

## 6. Conclusions

We have introduced LapSum, a new method that enhances the process of performing tasks such as ranking, sorting, and selecting the top-k elements from a dataset. It is straightforward theoretically, can be used for all soft-order tasks, and is differentiable with respect to all parameters. LapSum utilizes the properties of the Laplace distribution to improve both the speed and memory usage compared to other existing techniques. The effectiveness of the method has been supported by theoretical and experimental validation. Furthermore, we have attached an open source code for this approach, both in Python and in CUDA for execution on GPUs, making it accessible for broader community.

## Impact Statement

LapSum enables applications of soft ordering methods to large-scale real-world datasets. Consequently, LapSum has the potential to significantly impact industries relying on large-scale data processing, such as recommendation systems, natural language processing, and computer vision.

## Acknowledgments

This research was partially funded by the National Science Centre, Poland, grants no. 2020/39/D/ST6/01332 (work by Łukasz Struski) and 2023/49/B/ST6/01137 (work by Jacek Tabor). Some experiments were performed on servers purchased with funds from the flagship project entitled "Artificial Intelligence Computing Center Core Facility" from the DigiWorld Priority Research Area within the Excellence Initiative – Research University program at Jagiellonian University in Kraków.

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

# A. LapSum: pseudocode for solution of top-k problem

We present the complete pseudocode for the solution of top-w problem constructed in the framework of LapSum. For sequence $s = (s_i)_{i=0..n-1}$ we construct a sequence $p = (p_i) \in (0,1)^n$, such that

$$\sum_i p_i = k$$

and the operation $s \to p$ is differentiable. In the limiting case when the pairwise differences between elements of the sequence $s_i$ are large, the solution tends to hard solution to top-k problem.

We put

$$Lap(x) = \begin{cases} \frac{1}{2}\exp(x) & \text{for } x \leq 0, \\ 1 - \frac{1}{2}\exp(-x) & \text{otherwise.} \end{cases}$$

---

**Algorithm 1** LapSum: soft top-$k$ function
for the input $s = (s_i)_{i=0..n-1}$

---

**Require:** Sequence $(s_i)_{i=0..n-1}$, parameter $\in (0,n)$
1: Sort $s$ in decreasing into $r = (r_i)_{i=0..n-1}$
2: Set $r_{-1} = \infty$, $r_n = -\infty$
3: Initialize: $a_{n-1} = 0$, $b_0 = 0$, $c_0 = 0$
4: **for** $j = n-1$ to $0$ **do**
5: $\quad a_{j-1} = (1 + a_j) \cdot \exp(r_j - r_{j-1})$
6: **end for**
7: **for** $j = 0$ to $n-1$ **do**
8: $\quad b_{j+1} = (1 + b_j) \cdot \exp(r_{j+1} - r_j)$
9: $\quad c_{j+1} = 1 + c_j$
10: **end for**
11: Set $w_{-1} = 0$, $w_n = n$
12: **for** $j = 0$ to $n-1$ **do**
13: $\quad w_j = \frac{1}{2}a_j\exp(r_{j+1} - r_j) - \frac{1}{2}b_{j+1} + c_{j+1}$
14: **end for**
15: Find $j \in \{0, \dots, n\}$ such that $k \in [w_{j-1}, w_j]$
16: **if** $j = 0$ **then**
17: $\quad b = r_0 - \log 2 - \log k + \log a_0$
18: **else if** $0 < j < n$ **then**
$\quad b = r_{j+1} + \log a_j$
19:
$\quad - \log\left(k - c_{j+1} + \sqrt{|k - c_{j+1}|^2 + a_j b_{j+1} e^{r_{j+1} - r_j}}\right)$
20: **else if** $k = n$ **then**
21: $\quad b = r_{n-1} + \log 2 + \log(c_n - k) - \log b_n$
22: **end if**
23: **for** $i = 0$ to $n-1$ **do**
24: $\quad p_i = Lap(s_i - b)$
25: **end for**
26: **return** $p = (p_i)$

---

# B. Limiting case

**Theorem B.1.** *For every $k \in (0,n)$ and $r \in \mathbb{R}^n$ we have*

$$F\text{-Top}_\alpha(r,k) \in \Delta_k. \tag{7}$$

*Moreover, if $k$ is integer and the elements of $r$ are pairwise distinct, then*

- *$F\text{-Top}_\alpha(r,k) \to \text{top}\min_k(r)$ as $\alpha \to 0^+$,*
- *$F\text{-Top}_\alpha(r,k) \to \text{top}\max_k(r)$ as $\alpha \to 0_-$.*

*Proof.* Observe that (7) follows directly from the fact that $b = F\text{-Sum}_\alpha^{-1}(r,k)$, and thus $\sum_i F_\alpha(b - r_i) = k$.

We proceed to the proof of the limiting case, where we consider the case $\alpha \to 0^+$ (the case $\alpha \to 0^-$ can be shown analogously). Since the function $F\text{-Sum}_\alpha$ is equivariant, without loss of generality we can assume that the sequence $(r_i)$ is strictly increasing. Consider the values $b_\alpha = F\text{-Sum}_\alpha^{-1}(r,k)$. We show that

$$F_\alpha(b_\alpha - r_j) \to 0 \text{ as } \alpha \to 0^+. \tag{8}$$

If this were not the case then, trivially, there would exist a sequence $\alpha_m \to 0^+$ and $\varepsilon > 0$ such that

$$F_{\alpha_m}(b_{\alpha_m} - r_j) = F\left(\frac{b_{\alpha_m} - r_j}{\alpha}\right) \geq \varepsilon \text{ for all } m.$$

By continuity and fact that $F$ is strictly increasing, this implies that $\frac{b_{\alpha_m} - r_j}{\alpha} \geq c = F^{-1}(\varepsilon)$. Consequently for $i < k$, since $r_j - r_i > 0$, we would get

$$F_{\alpha_m}(b_{\alpha_m} - r_i) = F\left(\frac{b_{\alpha_m} - r_i}{\alpha_m}\right) = F\left(\frac{b_{\alpha_m} - r_j}{\alpha_m} + \frac{r_j - r_i}{\alpha_m}\right)$$
$$\geq F\left(c + \frac{r_j - r_i}{\alpha_m}\right) \to 1 \text{ as } m \to \infty$$

which would lead to the inequality

$$\lim_{m \to \infty} \sum_{i=0}^{k-1} F_{\alpha_m}(b_{\alpha_m} - r_i) + F_{\alpha_l}(b_{\alpha_m} - r_j) \geq k + \varepsilon > k.$$

This we have obtained a contradiction with the equality $\sum_{i=0}^{n-1} F_\alpha(b - r_i) = k$. This proves the assertion, since we trivially obtain that $F_\alpha(b - r_j) \to 0$ for $\alpha \to 0^+$ and $j \geq k$. Consequently, making use of equality $\sum_i F_\alpha(b - r_i) = k$ we obtain that $F_\alpha(b - r_j) \to 1$ for $\alpha \to 0^+$ and $j < k$. $\square$

# C. Calculation of derivatives

In this section we are going to obtain main formulas for derivatives. We show, in particular, that all derivatives can be easily directly computed. Moreover, what is important for the gradient optimization process the multiplication of all the derivatives on a vector can be computed in $O(n \log n)$ time.

We shall start with establishing some consistent notation. We are given $r = (r_i)_{i=0..n-1} \in \mathbb{R}^n$, $k \in (0,n)$ and $\alpha \neq 0$.

We consider the function

$$b = b(r, k, \alpha) = Lap\text{-Sum}^{-1}(r, k; \alpha)$$

which by definition satisfies the condition

$$\sum_i Lap_\alpha(b - r_i) = k, \tag{9}$$

where

$$Lap_\alpha(x) = \begin{cases} \frac{1}{2}\exp(x/\alpha) & \text{if } x/\alpha \leq 0, \\ 1 - \frac{1}{2}\exp(-x/\alpha) & \text{if } x/\alpha > 0. \end{cases}$$

We additionally consider the function

$$lap_\alpha(x) = \frac{1}{2\alpha}\exp(-|x/\alpha|),$$

which, for $\alpha > 0$, corresponds to the density of the Laplace distribution.

We put

$$p_i = Lap_\alpha(x - r_i) \in [0, 1],$$

$$s_i = lap_\alpha(b(r, k) - r_i) = \frac{1}{2\alpha}\exp(-|\frac{x-r_i}{\alpha}|)$$

$$= \frac{1}{\alpha}\min(p_i, 1 - p_i).$$

We will need the following notation:

$$S = S_\alpha(w) = \sum_i s_i,$$

$$q = \frac{1}{S}(s_0, \ldots, s_{n-1}) =$$

$$= \text{softmax}\left(-|\frac{x-r_0}{\alpha}|, \ldots, -|\frac{x-r_{n-1}}{\alpha}|\right).$$

After establishing the necessary notation we proceed to computation of the derivatives.

### C.1. Derivative of inverse of $Lap$-Sum

**Derivative of $Lap$-Sum$^{-1}$ with respect to $k$**   Differentiating (9) with respect to $k$, we get

$$\sum_j lap_\alpha(b(r, w; \alpha) - r_j) \cdot \frac{\partial b}{\partial w} = 1,$$

which leads to

$$\frac{\partial b}{\partial k} = \frac{1}{\sum_j lap_\alpha(b(r, k) - r_j)} = \frac{1}{S}. \tag{10}$$

**Derivative of $Lap$-Sum$^{-1}$ with respect to $\alpha$**   Differentiating (9) with respect to $\alpha$, we get:

$$\sum_i lap_\alpha(b - r_i)\frac{\frac{\partial b}{\partial \alpha}\alpha - (b - r_i)}{\alpha} = 0.$$

As a result,

$$\frac{\partial b}{\partial \alpha} = \frac{1}{\alpha}\frac{\sum_i lap_\alpha\left(\frac{b-r_i}{\alpha}\right)(b - r_i)}{\sum_i lap_\alpha\left(\frac{b-r_i}{\alpha}\right)}$$

$$= \frac{1}{\alpha}(b - \langle q, r\rangle).$$

**Partial derivatives of $Lap$-Sum$^{-1}$ with respect to $r$**
Differentiating (9) with respect to $r_i$, we obtain:

$$\sum_j lap_\alpha(b(r, k)) \cdot \left(\frac{\partial b}{\partial r_i} - 1\right) = 0,$$

and consequently:

$$\frac{\partial b}{\partial r_i}\sum_j lap_\alpha(b(r, k) - r_j) = lap_\alpha(b(r, k) - r_i),$$

which simplifies to:

$$\frac{\partial b}{\partial r_i} = \frac{lap_\alpha(b(r, k) - r_i)}{\sum_j lap_\alpha(b(r, k) - r_j)} = q_i,$$

or equivalently:

$$\frac{db}{dr} = q^T.$$

### C.2. Derivatives of $Lap$-Top

We put

$$p_i(r, w; \alpha) = Lap_\alpha(b - r_i) \text{ where } b = Lap\text{-Sum}^{-1}(r, k; \alpha).$$

Observe that $(p_i)$ is exactly the solution given by LapSum to soft top-k problems.

**Derivatives of $p_i(r_1, \ldots, r_n, k)$ with respect to $k$**   We have:

$$\frac{\partial p_i}{\partial k} = \frac{\partial}{\partial k}Lap_\alpha(b(r, k) - r_i) = lap_\alpha(b(r, k) - r_i) \cdot \frac{\partial b}{\partial k}.$$

Substituting into (10), we get:

$$\frac{\partial p_i}{\partial w} = \frac{lap_\alpha(b(r, k) - r_i)}{\sum_j lap_\alpha(x_j - b(x, k))}.$$

Thus:

$$\frac{dp}{dk} = q.$$

**Derivative of $p(\cdot, k)$ with respect to $\alpha$**   We have:

$$\frac{\partial p_i}{\partial \alpha} = \frac{\partial}{\partial \alpha}Lap_\alpha(b(r, k) - r_i) = \frac{\partial}{\partial \alpha}Lap\left(\frac{b - r_i}{\alpha}\right)$$

$$= lap_\alpha(b(r, k) - r_i) \cdot \left[\frac{\partial b}{\partial \alpha} - \frac{b - r_i}{\alpha}\right].$$

Substituting:

$$\frac{\partial p_i}{\partial \alpha} = lap_\alpha(b(r, k) - r_i) \cdot \left(\frac{b}{\alpha} - \frac{1}{\alpha}\langle q, r\rangle - \frac{b}{\alpha} + \frac{r_i}{\alpha}\right)$$

$$= \frac{s_i}{\alpha} \cdot (r_i - \langle q, r\rangle).$$

Thus:

$$\frac{\partial p}{\partial \alpha} = \frac{1}{\alpha}(s \odot r - \langle q, r\rangle s). \tag{11}$$

**Derivative $D$ of $p(\cdot, k)$ with respect to $r$** For $D = [d_{ij}]$, we have:

$$d_{ij} = \frac{\partial p_j}{\partial r_i} = lap_\alpha(b(r,k) - r_i) \cdot \left( \frac{\partial b}{\partial r_i} - \delta_{ij} \right).$$

Thus (in column vector notation):

$$D = [lap_\alpha(b - r_i)] \left[ \frac{\partial b}{\partial r_i} \right]^T - \text{diag}(lap_\alpha(b - r_i))$$

$$= s\, q^T - \text{diag}(s).$$

**Derivative of $p(\cdot, k)$ in the direction $v$** Since we want to compute without declaration of all matrix, we get the derivative in the direction of $v$:

$$D \circ v = [s\, q^T - \text{diag}(s)]\, v = \langle q, v \rangle\, s - s \odot v,$$

where $\odot$ denotes componentwise multiplication.

**Computing $v^T \cdot D$** We have

$$v^T \circ D = v^T \circ [s\, q^T - \text{diag}(s)] = \langle s, v \rangle\, q^T - s^T \odot v^T.$$

### C.3. Computing derivatives for the function $u = \log p$

In some cases, especially for CE loss, we will rather need $\log p$ instead of $p$.

**Value of $\log p$** We have:

$$\log p_i = g\left( \frac{b - r_i}{\alpha} \right),$$

where:

$$g(x) = \begin{cases} -\log 2 + \log(2 - \exp(-|x|)) & \text{for } x \geq 0, \\ -\log 2 + x & \text{for } x < 0. \end{cases}$$

To calculate derivatives, we need the auxiliary function:

$$h(x) = \begin{cases} 1 & \text{for } x \leq \frac{1}{2}, \\ \frac{1}{x} - 1 & \text{for } x > \frac{1}{2}. \end{cases}$$

**Derivative with respect to $k$** We have

$$\frac{\partial u_i}{\partial k} = \frac{1}{p_i} \cdot \frac{\partial p_i}{\partial k} = \frac{q_i}{p_i}.$$

Thus

$$\frac{\partial u_i}{\partial k} = q/p = h(p)/(\alpha\, S).$$

**Derivative of $u$ with respect to $\alpha$** We have:

$$\frac{\partial u}{\partial \alpha} = \frac{1}{p} \odot \frac{\partial p}{\partial \alpha}.$$

Substituting:

$$\frac{\partial u}{\partial \alpha} = \frac{1}{p} \odot \frac{1}{\alpha}(s \odot r - \langle q, r \rangle\, s).$$

Thus:

$$\frac{\partial u}{\partial \alpha} = \frac{1}{\alpha^2}(h(p) \odot r - \langle q, r \rangle\, h(p)).$$

**Derivative of $u$ with respect to $r$** We have

$$Du = \text{diag}\left( \frac{1}{p} \right) Dp = \text{diag}\left( \frac{1}{p} \right)(s\, q^T - \text{diag}(s))$$

$$= \frac{1}{\alpha}\left( h(p)q^T - \text{diag}(h(p)) \right)$$

**$Duv$ – Derivative with respect to $r$ in the direction $v$**

$$\frac{du}{dr}(v) = \frac{1}{\alpha}\left( \langle q, v \rangle \cdot h(p) - v \odot h(p) \right),$$

where $\odot$ denotes componentwise multiplication.

**Dual case: computation of $v^T Du$** We have

$$v^T \cdot Du = \frac{1}{\alpha}(\langle v, h(p) \rangle q^T - v^T \odot h(p)^T).$$

### C.4. Derivatives of
$(k_i)_{i=0..L-1} \to (Lap\text{-}Sum_\alpha^{-1}(r, k_i))_{i=0..L-1}$

We shall now consider the case when the input is $k = (k_i) \in \mathbb{R}^L$. This happens in particular in LapSum for soft sorting or the computation of soft permutation matrix.

We put

$$b_i = b_i(r, k_i; \alpha) = Lap\text{-}Sum_\alpha^{-1}(r, k_i)$$

and

$$B = B(r, k; \alpha) = (b_i)_{i=0..K-1} \in \mathbb{R}^L.$$

Our aim is to compute the derivatives of the function $B$ with respect to $w$, $r$ and $\alpha$. We apply here the formulas from Appendix C.1.

We first introduce matrix $Q$ which is defined as

$$Q = \begin{bmatrix} q_0^T \\ \vdots \\ q_{L-1}^T \end{bmatrix},$$

where $q_m \in \mathbb{R}^n$ is the (column) vector defined by

$$q_m = \text{softmax}(-\left| \tfrac{b_m - r_0}{\alpha} \right|, \ldots, -\left| \tfrac{b_m - r_{N-1}}{\alpha} \right|) \in \mathbb{R}^n.$$

**Derivative over $\alpha$** We have

$$\frac{\partial b}{\partial \alpha} = \frac{1}{\alpha}(b - Q \cdot r).$$

**Derivative over $w$** We have

$$\frac{\partial B}{\partial k} = \text{diag}((1/S_m)_{m=0..L-1}),$$

where

$$S_m = \frac{1}{2\alpha} \sum_i \exp(-\left| \tfrac{b_m - r_i}{\alpha} \right|)$$

**Derivative over $r$**  We have

$$\frac{db}{dr} = Q.$$

**Fast computation**  Our aim is to show how one can efficiently compute the left and right multiplication of the above derivatives by vectors. We present here the general idea for fast computation of all $(S_l)$, the other calculations can be done by applying a similar approach. For clarity of presentation, we restrict to the case $\alpha = 1$.

So assume that we want to efficiently compute the values

$$D_m = \sum_{i=0}^{n-1} \exp(-|b_l - r_i|)$$

for all $m$, that is, in a smaller complexity than $O(L \cdot n)$. As before, we assume that the sequences $r$ and $b$ are in an increasing order. Observe that

$$D_m = A_m + B_m,$$

where

$$A_m = \sum_{i:r_i \leq b_m} \exp(r_i - b_m), \ B_m = \sum_{i:r_i > b_m} \exp(b_m - r_i).$$

Let $n_m = \min\{i \in \{0, \ldots, n-1\} : r_i > b_m\}$ (and $n$ if there is no $i$ satisfying this condition). Then since the sequence $r$ is increasing we obtain that

$$A_m = \sum_{i=0}^{n_m-1} \exp(r_i - b_)), \ B_m = \sum_{i=n_m}^{n-1} \exp(b_m - r_i).$$

Clearly, since sequences $r$ and $b$ are increasing we obtain that

$$n_m \leq n_{m+1}.$$

Consequently

$$A_{m+1} = \exp(b_m - b_{m+1}) \cdot A_m + \sum_{i=n_m}^{n_{m+1}} \exp(r_i - b_{m+1})$$

and

$$B_m = \exp(b_m - b_{m+1}) \cdot B_{m+1} + \sum_{i=n_m}^{n_{m+1}} \exp(b_m - r_i).$$

Therefore, we can first compute $A_0$ and $B_{n-1}$, and compute the rest by the above iterative formulas. Finally, the complexity of the computations is $O(L + n)$ instead of $O(L \cdot n)$.

## D. Experiments

**Training from scratch on CIFAR100 and fine-tuning on ImageNet**  We adopt the identical experimental setup as (Petersen et al., 2022b) using the Adam optimizer (Kingma & Ba, 2015) for all models. A grid search was conducted to fine-tune both the $\alpha$ parameter of LapSum and the learn-

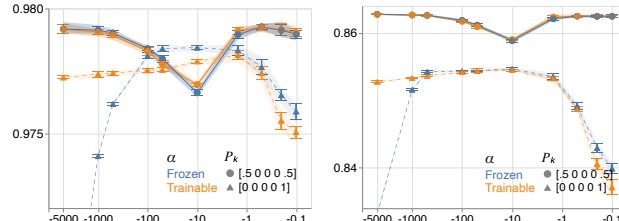

*Figure 8. ImageNet-1K* Top-1 and Top-5 accuracies of LapSum across varying values of $\alpha$ and $P_j$. Larger $\alpha$'s produce "harder", more discrete selections while smaller $\alpha$ lead to smoother outputs. This trade-off influences how strictly the model enforces a top-$k$ criterion

ing rates. We select $\alpha$ from $\{-1.5, -1, -0.5, -0.2, -0.1\}$ and Leaning Rate (LR) from $\{10^{-4.75}, 10^{-4.5}, 10^{-4.25}, 10^{-4}, 10^{-3.75}, 10^{-3.5}, 10^{-3.25}, 10^{-3.0}\}$, using early stopping approach too. See Fig. 8 for influence of $\alpha$ parameter on model performance. **Baselines** As baselines, we reuse scores reported in (Petersen et al., 2022b). **Implementation** We use code from https://github.com/Felix-Petersen/difftopk kindly made available by Petersen et al. (2022b). Hyperparameter details are available in Tab. 5.

*Table 5.* The hyperparameters for training CIFAR-100 from scratch and fine-tune of ImageNet-1K and ImageNet21K-P models.

|  | CIFAR-100 | ImageNet-1K | ImageNet-21K-P |
|---|---|---|---|
| Batch size | 100 | 500 | 500 |
| #Epochs | 200 | 100 | 40 |
| best LR | $10^{-3.25}$ | $10^{-4.25}$ | $10^{-4.75}$ |
| best $\alpha$ | -1.5 | -1.5 | -1.0 |
| Model | ResNet18 | ResNeXt-101 | ResNeXt-101 |

$k$**-NN for Image Classification**  We adopt an identical experimental setup as (Xie et al., 2020), which follows the approach of Grover et al. (2019). A grid search was conducted to fine-tune both the $\alpha$ parameter of LapSum and the learning rates. We set $\alpha$ as network trainable parameter. We select $\alpha$ at $t_0$ from $\{-10, -5, -3, -1.5, -1, -0.5, -0.2, -0.1\}$ and Leaning Rate (LR) from $\{10^{-4.75}, 10^{-4.5}, 10^{-4.25}, 10^{-4}, 10^{-3.75}, 10^{-3.5}, 10^{-3.25}, 10^{-3.0}, 10^{-2.75}\}$, also using early stopping. For details see Tab. 6. **Baselines** The baseline results are taken directly from Xie et al. (2020), which in turn references Grover et al. (2019) and He et al. (2016). **Implementation** We use the code kindly made available by Xie et al. (2020), which itself builds on Cuturi et al. (2019) and Grover et al. (2019).

**Trainable Parameter $\alpha$ in Soft-Permutation Experiment**  In our method, as described in the experiment in Sec. 5.2, we introduce a trainable parameter $\alpha$ that can

*Table 6.* $k$-NN experiments' hyperparameters.

| Dataset | MNIST | CIFAR-10 |
|---|---|---|
| $k$ | 9 | 9 |
| Batch size of query samples | 100 | 100 |
| Batch size of template samples | 100 | 100 |
| Optimizer | SGD | SGD |
| Number of epochs | 200 | 200 |
| Best LR | $10^{-2.75}$ | $10^{-3.25}$ |
| Best $\alpha$ | -1 | -10 |
| Momentum | 0.9 | 0.9 |
| Weight decay | $5 \times 10^{-4}$ | $5 \times 10^{-4}$ |
| Model | 2-layer CNN | ResNet18 |

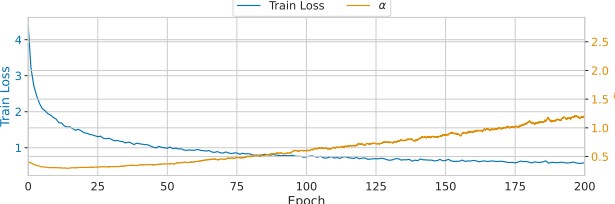

*Figure 9.* Results from one of the experiments in Sec. 5.2, showing the cost function (blue line) and the learning rate alpha (orange line) over training iterations. The cost function decreases as the model learns, indicating convergence. Parameter $\alpha$ starts at 0.4, initially decreases to allow for finer updates, and then increases to a value above 1 to accelerate learning in later stages.

be optimized on the basis of gradient during the training process. This parameter plays a crucial role in controlling the dynamics of the soft-permutation mechanism, allowing the model to adaptively balance between exploration and exploitation during learning. $\mathrm{LapSum}$ stays fully differentiable for all $\alpha \neq 0$.

Fig. 9 illustrates the evolution of $\alpha$ alongside the cost function over the course of training iterations. The blue line represents the cost function, which decreases smoothly and consistently, indicating stable convergence of the model. The orange line tracks the value of $\alpha$, which starts at an initial value of 0.4. Initially, $\alpha$ decreases to facilitate finer and more precise updates to the model parameters. As training progresses, $\alpha$ increases to a value greater than 1, allowing faster learning and greater updates at later stages of training. This adaptive behavior of $\alpha$ demonstrates its effectiveness in guiding the optimization process, ensuring both stability and efficiency in learning.

**Error, Runtime and Memory Analysis** We provide a detailed runtime and memory analysis of our method compared to other approaches. We use float64 precision whenever possible and float32 otherwise. Methods that support float64 precision are: LapSum (ours), Optimal Transport (Xie et al., 2020), SoftSort (Prillo & Eisenschlos, 2020), Fast, Differentiable and Sparse Top-k(Dykstra) (Sander

et al., 2023). Figs. 10 and 11 present the time complexity and memory usage of all the soft top-k methods considered when executed on a CPU. Similarly, Figs. 12 and 13 show the same metrics, but computed on a GPU. These figures illustrate the relationship between runtime, memory consumption, and data dimensionality $n$, with $k = n/2$. Across all scenarios, our approach (represented by the solid blue line) consistently outperforms or matches the best competing methods, particularly for higher data dimensions. Some competing algorithms fail to complete tasks due to excessive memory requirements, further highlighting the efficiency of our method. In addition, we include critical difference (CD) diagrams to statistically validate performance comparisons (Demšar, 2006). The CD diagrams confirm that our method ranks within the top-performing group of algorithms, demonstrating its superiority in both time and memory efficiency across CPU and GPU implementations. These results reinforce the robustness and scalability of our approach in practical applications.

The performance of soft permutation methods is visualized in Fig. 14 where the data dimension $n$ on the horizontal axis is compared to the memory usage and the computation time on the vertical axis. Computations were performed on a CPU due to the high memory demands of some of the considered methods. Results are shown for the forward process (left column) and the forward-backward process (right column). Our implementation is in PyTorch, while the other methods are implemented in TensorFlow. Our approach demonstrates superior scalability and efficiency, particularly for large $n$, achieving faster computation times and lower memory usage compared to the other methods.

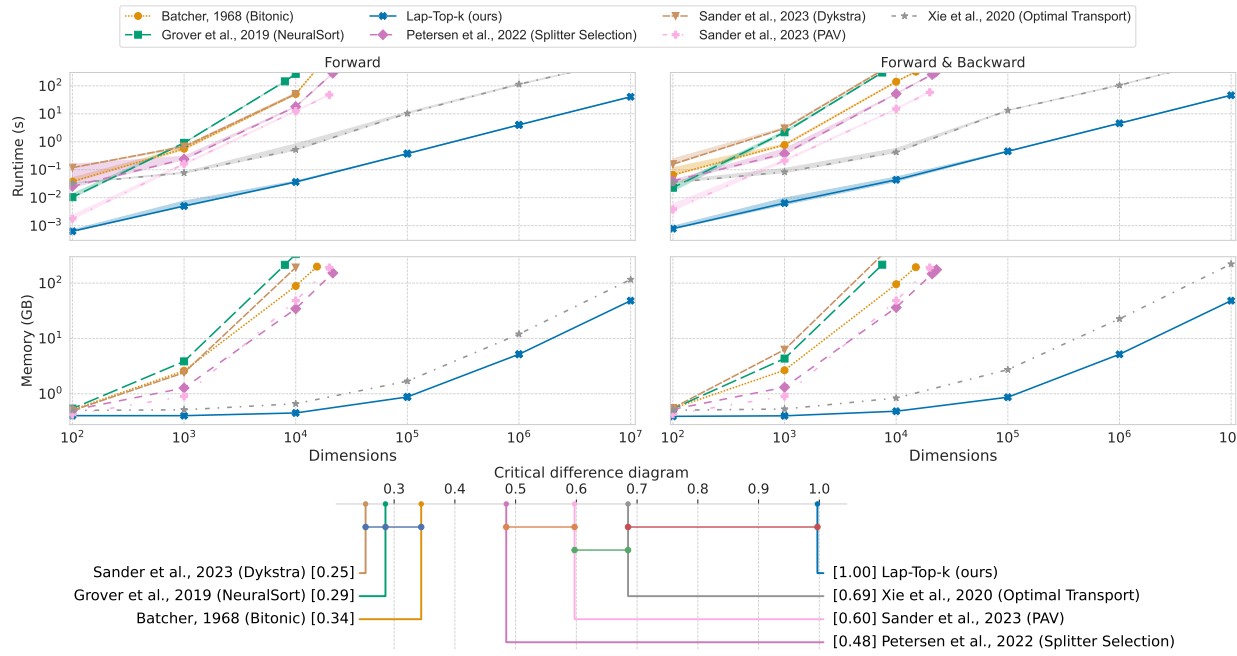

*Figure 10.* Relationship between the dimension ($n$), shown on the horizontal axis, and the maximum memory usage and computation time on vertical axis, for several functions with a fixed $k = 5$. All calculations displayed in this graph were performed on a CPU. The evaluation examined memory consumption and execution time during the forward pass (left column) and the combined forward and backward passes (right column).

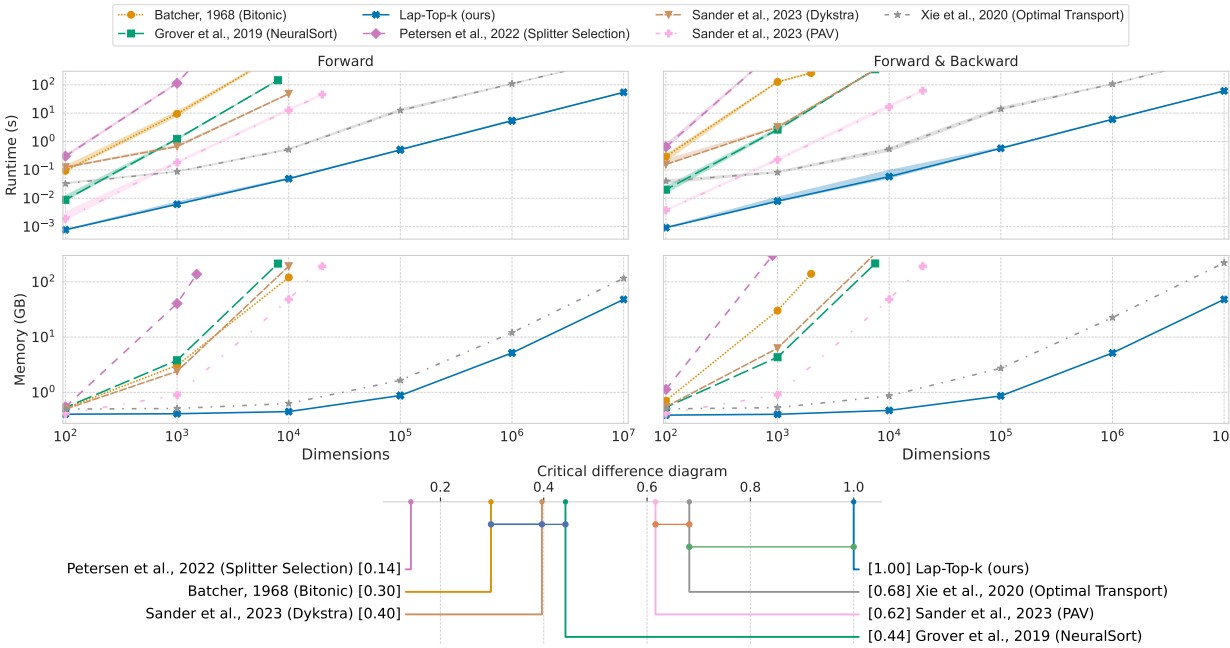

*Figure 11.* Relationship between the data dimension ($n$), represented on the horizontal axis, and the maximum memory usage and computation time, represented on the vertical axis. The parameter $k$ was dependent on $n$, calculated using the formula $k = n/2$. The computations depicted in this figure were performed on a CPU. The relationships between time and memory usage were analyzed during the forward process (charts in the left column) and the forward and backward processes (charts in the right column).

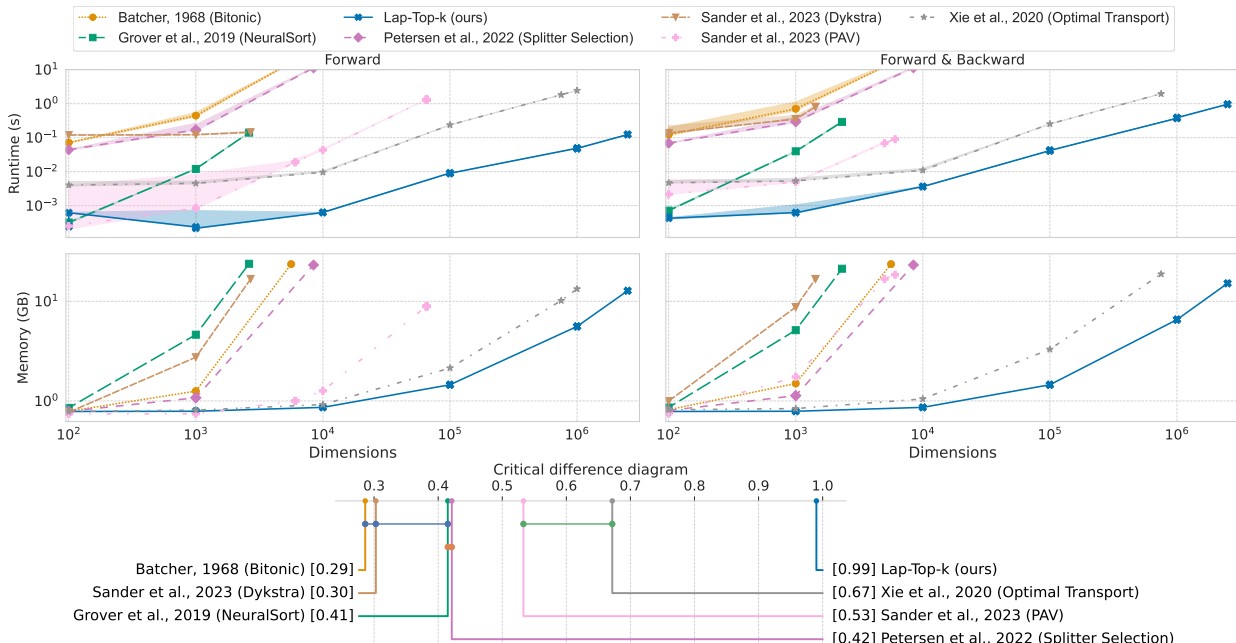

*Figure 12.* Relationship between the dimension $n$ (horizontal axis) and the maximum memory usage and computation time, represented on the vertical axis, for several functions with a fixed $k = 5$. All calculations displayed in this graph were performed on a GPU. The evaluation examined memory consumption and execution time during the forward pass (left column) and the combined forward and backward passes (right column).

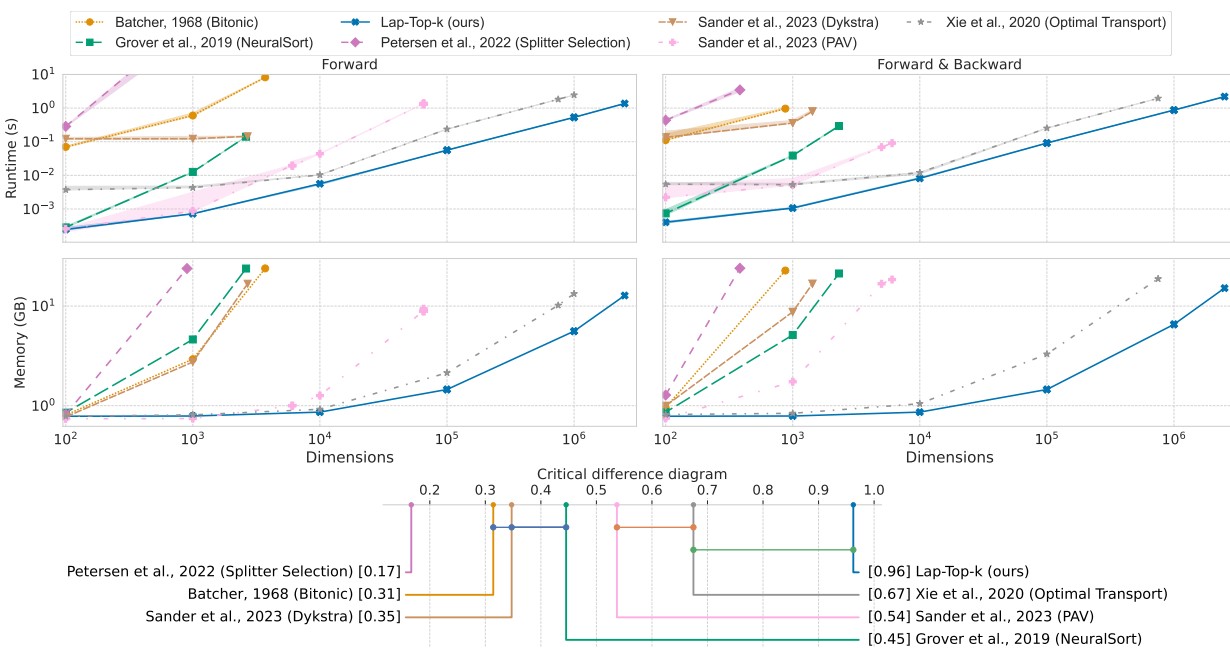

*Figure 13.* The figure illustrates the correlation between the dimensionality $n$ on horizontal axis and the peak memory usage as well as computation time, on vertical axis, for various functions with $k = n/2$. All computations depicted in this graph were executed on a GPU. The analysis focused on memory utilization and runtime during both the forward pass (left column) and the combined forward and backward passes (right column).

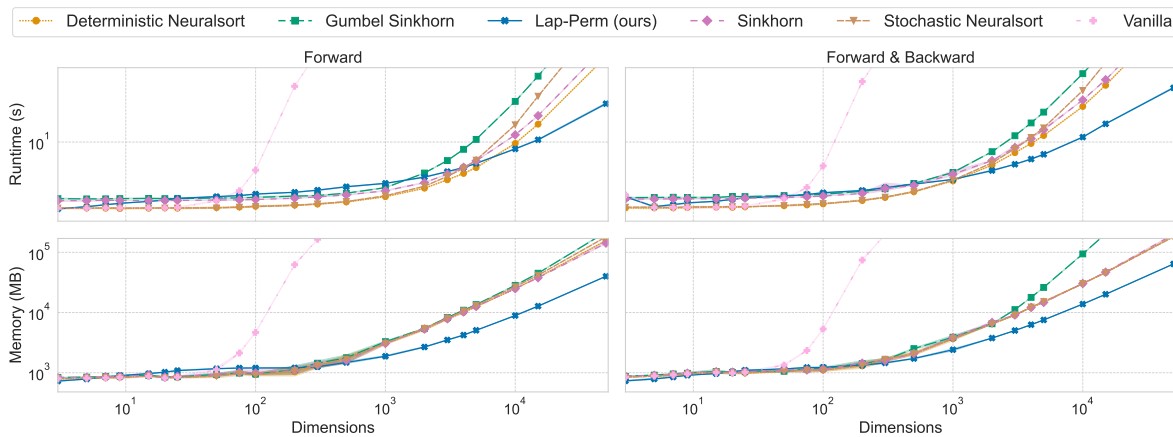

*Figure 14.* The performance of soft permutations methods, plotting data dimension $n$ on the horizontal axis versus memory usage and computation time on the vertical axis. Computations were performed on a CPU due to the high memory demands of some of the considered methods. Results are shown for the forward process (left column) and forward-backward process (right column). The implementation of our method is in PyTorch, while the others are implemented in TensorFlow. Our approach demonstrates superior scalability and efficiency, particularly for large $n$, achieving faster computation times and lower memory usage compared to the other methods.

