# OpenReview forum: "LapSum - One Method to Differentiate Them All: Ranking, Sorting and Top-k Selection"
_ICML.cc/2025/Conference — ICML 2025 poster_

### Official Review · Reviewer_m9fB · 2025-03-12

**Overall Recommendation:** 4

**Summary:**

LapSum introduces a unified method for creating differentiable versions of ordering operations—such as ranking, sorting, and top‑k selection—by leveraging a closed-form inversion of the Lap-Sum function (the sum of Laplace CDFs). This approach allows efficient gradient computation in $O(n \log n)$ time while using only $O(n)$ memory, making it well-suited for high-dimensional data. The theoretical framework shows that these soft approximations converge to their hard, discrete counterparts. Extensive experiments on datasets like CIFAR‑100 and ImageNet demonstrate competitive or superior performance compared to existing methods, and the availability of both CPU and CUDA implementations underscores its practical applicability in large-scale neural network training and optimization tasks.

**Claims And Evidence:**

I think the evidence is generally clear and convincing.

**Essential References Not Discussed:**

I am not familiar with literature at all.

**Experimental Designs Or Analyses:**

The experimental design is generally sound.

**Methods And Evaluation Criteria:**

The methods are well-justified for the problem at hand.

**Other Comments Or Suggestions:**

Overall, I find the paper both rigorous and engaging. However, for readers like me who may not be deeply familiar with the literature, it would be beneficial to include concrete examples that explain in greater detail the importance and applications of ranking, sorting, and top‑k selection. Additionally, incorporating a concise pseudocode or algorithmic summary that outlines the key steps of LapSum in the main text would help clarify the intuition behind the algorithm and its general use.

**Other Strengths And Weaknesses:**

- Strengths: From my perspective, I think this paper is well-written, and novel in the sense that this it proposes unified framework for differentiable ordering that creatively combines ideas from previous works on smooth approximations with a closed-form solution using the Lap-Sum function.

- Weakness: since I am not familiar with literate, experinments only focus on the calssification tasks, which might hurt the broad implicaition of the current method. Another application can be done to demonstrate the generality of the approach.

**Questions For Authors:**

See the comment above.

**Relation To Broader Scientific Literature:**

The paper situates its contributions within a growing body of work on differentiable ordering operations. Previous approaches often faced challenges in computational efficiency or relied on iterative procedures, LapSum introduces a closed-form inversion using the Lap-Sum function, leading to a unified framework that efficiently computes gradients in O(n log n) time and requires only O(n) memory.

**Theoretical Claims:**

I have not checked the proof, but theorem statements are reasonable for me.

---

> ### Author Rebuttal · Authors · 2025-03-31
>
> Thank you for your detailed review and constructive suggestions for improving our paper.
>
> 1. Applications of our method
>
>    Our paper introduces several applications including top-k learning, soft ranking, sorting, and permutation learning. To address your request for concrete examples:
>
>    - Vector Quantized-VAE models can directly leverage our differentiable sorting approach in their codebook optimization process, improving representation learning efficiency.
>
>    - Large Language Models benefit from our ranking method when analyzing token distributions and building optimized dictionaries, enabling more effective model analysis and interpretation. Finding the dictionary of a trained Transformer dictionary provides a method for its deeper analysis.
>
>    We have included pseudocode for the LapSum model in Appendix A, designed for straightforward implementation. Additionally, the attached code contains complete implementations in both PyTorch and CUDA, with performance competitive with existing methods.
>
> 2. Experiments
>
>    While our paper focused primarily on classification applications, we will expand the experimental section to include soft ranking and sorting evaluations. These additions will demonstrate that LapSum provides a unified approach applicable across all differentiable soft ordering problems. The results will show comparable or superior performance to specialized methods while maintaining our closed-form advantages. [Link to figures with comparison of ranking and sorting solutions.](https://anonymous.4open.science/r/icml25-7AB6/README.md)
>
> Results for sorting methods on forward (CPU, 10* - space dimension):
> |Metric|Method|10⁷|10⁶|10⁵|10⁴|10³|
> |:-|:-|-:|-:|-:|-:|-:|
> |Mean time|Lap-Sort|59.944|6.163|0.596|0.059|0.009|
> ||Blondel et al.|98.884|6.077|0.443|0.044|0.007|
> |Max memory(MB)|Lap-Sort|43.36|4.74|0.87|0.49|0.45|
> ||Blondel et al.|29.09|3.44|0.84|0.57|0.56|
>
> Results for ranking methods on forward (CPU, 10* - space dimension):
> |Metric|Method|10⁷|10⁶|10⁵|10⁴|10³|
> |:-|:-|-:|-:|-:|-:|-:|
> |Mean time|Lap-Rank|32.379|2.81|0.245|0.021|0.003|
> ||Blondel et al.|110.712|6.034|0.437|0.04|0.008|
> |Max memory|Lap-Rank|33.83|3.85|0.85|0.55|0.52|
> ||Blondel et al.|19.53|2.45|0.75|0.57|0.55|
>
> We appreciate your thoughtful feedback and believe these clarifications and additions will strengthen the paper considerably.

---

### Official Review · Reviewer_wMdT · 2025-03-13

**Overall Recommendation:** 4

**Summary:**

Authors propose “LapSum” - that yields differentiable versions of ranking, sorting, top-k selection, and permutations, all in closed form, with low time complexity: O(nlogn) (same as any sorting algorithm), and a linear memory.

Authors define this F-sum function and then define the ranking task in terms of F-sum function.

The main contributions is: 1) Defining all “soft” ordering tasks can be built by plugging the relevant r and appropriate $\alpha$ and then inverting or evaluating F-Sum(\alpha). A naive approach would need iterative solutions and result in O(n^2) complexity. 2) Authors choose
F as the CDF of the Laplace distribution, the computations become both closed-form and O(nlogn) complexity.

## update after rebuttal

I had one clarification question around experiments with learning-to-rank datasets, and authors clarified that it is not the norm in related works to experiment with learning-to-rank datasets, and authors experimental setup is indeed valid. My original assessment (4) has not changed.

**Claims And Evidence:**

The claims made hold theoretically, and empirically, though I have some reservations (more of that in questions to the authors section).

**Essential References Not Discussed:**

Authors didn't discuss the connection (or a lack of) with the following differentiable ranking works:

1. Ustimenko, Aleksei, and Liudmila Prokhorenkova. "Stochasticrank: Global optimization of scale-free discrete functions." International Conference on Machine Learning. PMLR, 2020.

2. Oosterhuis, Harrie. "Learning-to-rank at the speed of sampling: Plackett-luce gradient estimation with minimal computational complexity." Proceedings of the 45th International ACM SIGIR Conference on Research and Development in Information Retrieval. 2022.

3. Sakhi, Otmane, David Rohde, and Nicolas Chopin. "Fast slate policy optimization: Going beyond Plackett-Luce." arXiv preprint arXiv:2308.01566 (2023).

**Experimental Designs Or Analyses:**

I am not fully convinced with the experimental setup. I am open to discussion with the authors on this.

**Methods And Evaluation Criteria:**

Authors use benchmark datasets used in previous papers from the "differentiable sorting" literature. I have some additional comments (more of that in questions to the authors section).

**Other Comments Or Suggestions:**

As I noted in the previous section, if the authors can first formally introduce the ranking task (mathematically) and it's connection with CDFs and the math following, it would be more helpful for the readers.

**Other Strengths And Weaknesses:**

The writing of the paper could be improved, for example the connection of the CDF with the ranking task is not immediately clear. It would be helpful if authors first formally write down the ranking task (for ex: rank as sum of indicator function etc. to connect with a CDF).

Also, the method is designed for ranking tasks, but popular learning to rank datasets/tasks are missing, for ex: MSLR, Yahoo etc LTR datasets [1].


1. Ustimenko, Aleksei, and Liudmila Prokhorenkova. "Stochasticrank: Global optimization of scale-free discrete functions." International Conference on Machine Learning. PMLR, 2020.

**Questions For Authors:**

Please see my comments previously about the lack of connections to some previous works and lack of learning to rank datasets/task.

**Relation To Broader Scientific Literature:**

The paper proposes a model for learning to rank task, which is critical to many real world applications.

**Theoretical Claims:**

The theoretical claims seem to be correct, though I didn't get a chance to go through all of the steps in all proofs.

---

> ### Author Rebuttal · Authors · 2025-03-31
>
> Thank you for your thoughtful review of our paper.
>
> 1. Related references
>
>    We appreciate your suggestions regarding additional references. Our literature review focused primarily on differentiable soft ordering, ranking, sorting, and top-k methods as addressed in works by Cuturi, Lapin, Berrada, Petersen, Blondel, and others (all included in bibliography). While our primary aim was to propose a new theoretically well-founded and computationally efficient tool to compare with these approaches, we acknowledge the importance of gradient boosting stochastic smoothing methods and Plackett-Luce based models. We will include these references in our revised manuscript.
>
> 2. Experiments
>
>    For direct comparisons, we utilized applications addressed in existing literature, such as top-k classification and permutation matrix classification, particularly in high-dimensional spaces. While the additional datasets you suggest are very valuable, incorporating them would extend beyond our current scope. We plan to evaluate our framework on a wider range of applications in future work.
>
>    Regarding experimental setup, we followed protocols established in the aforementioned literature to ensure direct compatibility, as our main goal was to develop a well-grounded and computationally effective tool. The direct comparisons with boosting approaches are in plans for our research.
>
>    We have not included soft ranking due to limited availability of solutions with probability values. However, we have defined the LapSum approach for both ranking and sorting tasks and will include these results in the final version. [Link to figures with comparison of ranking and sorting solutions.](https://anonymous.4open.science/r/icml25-7AB6/README.md)
>
> Results for sorting methods on forward (CPU, 10* - space dimension):
> |Metric|Method|10⁷|10⁶|10⁵|10⁴|10³|
> |:-|:-|-:|-:|-:|-:|-:|
> |Mean time|Lap-Sort|59.944|6.163|0.596|0.059|0.009|
> ||Blondel et al.|98.884|6.077|0.443|0.044|0.007|
> |Max memory(MB)|Lap-Sort|43.36|4.74|0.87|0.49|0.45|
> ||Blondel et al.|29.09|3.44|0.84|0.57|0.56|
>
> Results for ranking methods on forward (CPU, 10* - space dimension):
> |Metric|Method|10⁷|10⁶|10⁵|10⁴|10³|
> |:-|:-|-:|-:|-:|-:|-:|
> |Mean time|Lap-Rank|32.379|2.81|0.245|0.021|0.003|
> ||Blondel et al.|110.712|6.034|0.437|0.04|0.008|
> |Max memory|Lap-Rank|33.83|3.85|0.85|0.55|0.52|
> ||Blondel et al.|19.53|2.45|0.75|0.57|0.55|
>
> 3. Theoretical claims
>
>    Thank you for recommending a more formal definition connecting ranking to the CDF. We will incorporate this description to improve clarity, following your recommended approach. A key advantage of LapSum is its theoretically elegant closed-form solution, naturally not forgetting its computational efficiency. This non-iterative approach provides enhanced stability compared to existing methods that rely on iterative processes.
>
> 4. Other concerns and concluding remarks
>
>    We believe we have addressed your primary concerns and will implement the suggested improvements in our final manuscript. In the revised version, we will add the missing references to better describe the related research field, include a formal description of ranking and CDF, and more clearly articulate the impact of our proposal on the field.

---

### Official Review · Reviewer_UBWD · 2025-03-24

**Overall Recommendation:** 3

**Summary:**

This paper proposes a new method for computing differentiable approximations of ranking, sorting and top-k operators. This method is based on considering sums of the CDF of the Laplace distribution, which defines the approximations for well chosen arguments, with a regularization term $\alpha$. The choice of the Laplace distribution is motivated by the fact that the proposed operators can be computed and differentiated in closed form efficiently in this case, as formally detailed in the paper.
The method is illustrated for top-k selection in multilabel classification on CIFAR and Imagenet, as well as k-nn for Image classification on MNIST and CIFAR and soft-permutation on MNIST.
Experimental results also validate the computational and memory efficiency of the proposed approach for the top-k operator, either surpassing or competing with previous approximations on different hardware.

**Claims And Evidence:**

- the efficient calculation of the Lap-Sum, its inverse and derivatives are supported by convincing mathematical evidence. I nevertheless think it would have been clearer to formalize the results of paragraph "Calculation of inverse function" (l.269) and section 4.2 (derivatives) with a proposition, as is done for section 4.1.

- the fact that $F-Rank_\alpha$ and $F-Top_\alpha$ approximate the ranking and top-k operator is supported by convincing mathematical evidence. I would have expected the same for $F-Sort_\alpha$, for which, unless I am mistaken, a proof is missing.

**Essential References Not Discussed:**

Most papers I know about in the differentiable programming area to propose relaxations of sorting, ranking, or top-k operators are discussed, except Berthet et al., 2020: Learning with Differentiable Perturbed Optimizers, which is clearly aligned with the topic considered, by proposing soft ranking operators for label ranking applications.

**Experimental Designs Or Analyses:**

- Yes, I checked the soundness/validity of the experimental designs, which are mostly adapted from previous works. I did not notice any issue despite the aforementioned one, which consists of missing experiments for the sorting and ranking operators.

- I appreciated the empirical validation of the efficiency of the proposed method for top-k selection.

- Displaying the std for the results in the tables would have been helpful to validate the statistical significance of the results.

**Methods And Evaluation Criteria:**

The paper considers evaluations and methods that were used in prior works, such as Petersen et al. (2022b), Berrada et al. (2018), and Grover et al. (2019). The evaluations mostly concern the top-k operator, with also one experiment on soft-permutations.

To me, the evaluation makes sense for the problem and application at hand, but it would have strengthened the paper to add applications on sorting and ranking, such as the ones in Blondel et al., 2020, or in Berthet et al., 2020 (Learning with Differentiable Perturbed Optimizers).

**Other Comments Or Suggestions:**

I strongly suggest the authors carefully proofread the paper to eliminate the typos. Here is a non-exhaustive list:

- l. 26: optimal transport-based?
- l. 29
- l. 33: what are n and k?
- "Techniques employed to solve these problems include relaxations and estimators, ranking regularizers, even learning-based ranking solutions": add references.
- l. 69: "value of k" + parentheses for the ref.
- l. 117: I find it misleading that F_alpha and f_alpha correspond to two different formulas, but this is a detail.
- l. 154: inconsistency between the definition of F_rank here and in thm. 3.2 (in terms of argument, scalar vs. vector).
- l. 170
- l. 215: doubly stochastic?
- l. 270: the function?
- l. 305: Appendix x2
- l. 410

**Other Strengths And Weaknesses:**

Strengths:

- The paper proposes a method that is faster than existing ones for computing and differentiating differentiable approximations of the top-k operator.

- The method is easy to understand and to implement.

I tend to lean towards acceptance of the paper, but there are a few weaknesses that bother me:

- The paper introduces differentiable approximations for sorting and ranking, but these are not considered in the experimental part of the paper.

- The sentence "Through extensive experiments, we demonstrate that our method outperforms state-of-the-art techniques for high-dimensional vectors and large k values" feels like an overstatement to me. It is true that the proposed method outperforms existing ones in terms of speed and memory (even though I did not see any experimental analysis for sorting and ranking), but it does not demonstrate superiority in terms of accuracy.

- There are too many typos.

- The method is really comparable to Xie et al. (2020) in terms of efficiency: can the authors comment on the advantage of their method compared to this one? Why should we use one instead of the other?

- The proposed top-k operator is not sparse (in the sense that as soon as $\alpha \neq 0$, all the coefficients will be non zero). This should be mentioned because this prevents the use of the operator for pruning weights in neural networks or for mixture of experts.

**Questions For Authors:**

- Why haven't you considered applications to sorting and ranking ?

- Lap Sum is really comparable to Xie et al (2020) in terms of computational and memory efficiency: can the authors comment on the advantage of their method compared to this one ? Why should we use one instead of the other?

**Relation To Broader Scientific Literature:**

The key contributions of the paper are quite well related to the broader scientific literature. The paper clearly cites existing papers proposing competing methods, and compares itself to these methods in terms of accuracy (top-k classification, soft-permutation, k-nn in tab 1-4) as well as efficiency (space and time complexity).

However, I still think that a comparison is missing for sorting and ranking, since the experimental part of the paper solely focuses on top-k and soft-permutations.

**Theoretical Claims:**

I checked the proofs in the main text, which seem okay to me despite some typos.

- The fact that F-Rank_alpha is applied at $r_k$ in l.154 versus applied on $r$ in theorem 3.2 is confusing. I understand that l.154 should be $(F-Rank_\alpha(r))_k$ instead?
- For the proof of theorem 3.2, can the authors clarify why there is a 1/2 factor which we don't have in the theorem?
- The definition of F-sort_alpha involves the inverse of F-sum_alpha, but this one is only evaluated with one argument, whereas it expects two arguments (see l.125, for instance).
- Typo in the definition of F-Top_alpha.

---

> ### Author Rebuttal · Authors · 2025-03-31
>
> Thank you for your very careful reading and insightful review of our paper. We address your questions as follows:
>
> 1. Why haven't we considered sorting and ranking applications?
>
>    Our research focused on designing an efficient model with a closed form for soft ordering tasks, and we defined models for both ranking and sorting. For applications, we deliberately aligned with Petersen's established framework to facilitate meaningful comparisons. Our experimental scope was partially constrained by implementations that allow cross-entropy loss value comparisons. Nevertheless, we conducted experiments for both sorting and ranking using LapSum, achieving results comparable to existing solutions. These will be included in the final version. [Link to figures with comparison of ranking and sorting solutions.](https://anonymous.4open.science/r/icml25-7AB6/README.md)
>
> Results for sorting methods on forward (CPU, 10* - space dimension):
> |Metric|Method|10⁷|10⁶|10⁵|10⁴|10³|
> |:-|:-|-:|-:|-:|-:|-:|
> |Mean time|Lap-Sort|59.944|6.163|0.596|0.059|0.009|
> ||Blondel et al.|98.884|6.077|0.443|0.044|0.007|
> |Max memory(MB)|Lap-Sort|43.36|4.74|0.87|0.49|0.45|
> ||Blondel et al.|29.09|3.44|0.84|0.57|0.56|
>
> Results for ranking methods on forward (CPU, 10* - space dimension):
> |Metric|Method|10⁷|10⁶|10⁵|10⁴|10³|
> |:-|:-|-:|-:|-:|-:|-:|
> |Mean time|Lap-Rank|32.379|2.81|0.245|0.021|0.003|
> ||Blondel et al.|110.712|6.034|0.437|0.04|0.008|
> |Max memory|Lap-Rank|33.83|3.85|0.85|0.55|0.52|
> ||Blondel et al.|19.53|2.45|0.75|0.57|0.55|
>
> 2. Justification for using our model over Xie's given comparable computation time and memory
>
>    While both models demonstrate similar computational efficiency as shown in our statistical tests (Figures 5, 7), LapSum outperforms Xie's model in all but one test. Both methods offer high accuracy (important in application, as Xie notes) and handle all derivatives effectively. However, the key advantage of our approach is its closed-form solution, whereas Xie's method relies on an iterative approach for both the model and its derivatives. This closed-form formulation provides theoretical elegance and potentially better stability in complex applications.
>
> 3. Missing reference to Berthet's paper
>
>    We acknowledge the omission of Berthet et al. 2020 "Learning with differentiable..." paper. While we referenced a Blondel et al. 2020 "Fast differentiable..." paper which addresses related solutions, we will include Berthet's work in our final version, as it aligns very well with our approach and provides excellent context on related research.
>
> 4. Theoretical claims
>
>    Thank you for your detailed analysis. We confirm that:
>    - The definition in line 154 should indeed be $(F-Rank_\alpha(r))_k$ and will be corrected. In later lines, where some indices may be obvious, we shall note it clearly.
>    - For $F-Sort_\alpha$ (line 125), we will add a footnote clarifying that $r$ in the definition is the default parameter.
>    - In Theorem 3.2 proof the ${1}\over{2}$ is the value of $F_\alpha$ function at $0$, see e.g. Fig.2. The mismatch between the formulation and proof of the theorem is a typo to be corrected.
>    - The proof for $F-Sort_\alpha$ is very similar, and we can add it in the final version.
>
> 5. Experimental issues
>
>    We appreciate your positive assessment of our experimental flow. Regarding standard deviations in experiment tables: we relied on published results from other researchers where original code wasn't always available. Reimplementing these approaches would introduce potential inconsistencies and errors. Consequently, we cannot provide deviation values at this stage but acknowledge this as an area for future work.
>
> 6. Potential overstatement in abstract
>
>    Our intention was to emphasize that our tool is theoretically sound, offers a closed-form solution, and is computationally competitive for high-dimensional spaces. The wording was not meant to imply superiority but rather computational competitiveness. We will revise this to better reflect our position.
>
> 7. Typographical errors
>
>    We will correct all identified typos (as addressed above) to improve clarity. Regarding "doubly stochastic matrix" we were referring to permutation matrices $P_{k,c}$ that are both row- and column-stochastic, as described in Petersen 2022 "Differentiable top-k..." The $n$ and $k$ in the introduction denote the space dimension and the number of top values to be selected — this shall be clarified. Thank you for careful reading.

---

### Decision · Program_Chairs · 2025-05-01

**Decision:**

Accept (poster)

**Comment:**

This paper presents LapSum, a method for differentiable ranking, sorting, and top‑k selection using a closed-form inversion of a Laplace-based sum. It offers memory-efficient gradient computation and applies to high-dimensional settings. The approach is theoretically sound, and scales well on both CPU and GPU. While experiments focus mainly on top‑k tasks, the authors provide additional results and definitions for sorting and ranking.

I recommend acceptance. The method is original. Its closed-form nature sets it apart from prior iterative approaches. Despite minor presentation issues, the authors addressed reviewer concerns and clarified their contributions. This work is a valuable addition to differentiable optimization research.